# The Wnt/TCF7L1 transcriptional repressor axis drives primitive endoderm formation by antagonizing naive and formative pluripotency

Paraskevi Athanasouli [1,10], Martina Balli[1,10], Anchel De Jaime-Soguero [1] ✉, Annekatrien Boel[2], Sofia Papanikolaou[3,4], Bernard K. van der Veer[1], Adrian Janiszewski[1], Tijs Vanhessche[1], Annick Francis[5], Youssef El Laithy[1], Antonio Lo Nigro [1], Francesco Aulicino [6], Kian Peng Koh[1], Vincent Pasque [1,7], Maria Pia Cosma [6,8,9], Catherine Verfaillie [1], An Zwijsen [5], Björn Heindryckx [2], Christoforos Nikolaou[4] & Frederic Lluis [1] ✉

Early during preimplantation development and in heterogeneous mouse embryonic stem cells (mESC) culture, pluripotent cells are specified towards either the primed epiblast or the primitive endoderm (PE) lineage. Canonical Wnt signaling is crucial for safeguarding naive pluripotency and embryo implantation, yet the role and relevance of canonical Wnt inhibition during early mammalian development remains unknown. Here, we demonstrate that transcriptional repression exerted by Wnt/TCF7L1 promotes PE differentiation of mESCs and in preimplantation inner cell mass. Time-series RNA sequencing and promoter occupancy data reveal that TCF7L1 binds and represses genes encoding essential naive pluripotency factors and indispensable regulators of the formative pluripotency program, including *Otx2* and *Lef1*. Consequently, TCF7L1 promotes pluripotency exit and suppresses epiblast lineage formation, thereby driving cells into PE specification. Conversely, TCF7L1 is required for PE specification as deletion of *Tcf7l1* abrogates PE differentiation without restraining epiblast priming. Taken together, our study underscores the importance of transcriptional Wnt inhibition in regulating lineage specification in ESCs and preimplantation embryo development as well as identifies TCF7L1 as key regulator of this process.

In early preimplantation development, the inner cell mass (ICM) cells face a binary decision between two distinct cell lineages: the naive pluripotent epiblast (EPI), which will form the embryo proper, and the extra-embryonic primitive endoderm (PE) cells, which will give rise to the endodermal component of the visceral and parietal yolk sac. On mouse embryonic day (E) 4.5, EPI and PE lineages are fully segregated, forming the expanded blastocyst, which is then ready for implantation in the uterus[1–3]. Genetic studies have shown that the established EPI cell fate is centered on the transcriptional network mastered by NANOG, while PE fate is governed by GATA6 expression[4–6].

mESCs are the in vitro counterpart of the pluripotent pre-implantation EPI and can be propagated indefinitely in pluripotent culture conditions[7]. However, upon removal of self-renewing

conditions, mESCs differentiate into post-implantation EPI-primed cells by first transiting through a formative pluripotent state[8,9]. During this process, naive pluripotency transcription factors (TFs) such as *Nanog* and *Prdm14* are downregulated while formative pluripotency regulators such as *Lef1* and *Otx2* are induced together with the de novo methyltransferases *Dnmt3a* and *Dnmt3b*[10,11]. Although mESCs resemble the preimplantation EPI, they can also spontaneously transit towards an extraembryonic PE-like state when cultured in naive pluripotency conditions[12,13], or by overexpression of the PE-specific genes *Gata6, Gata4* or *Sox17*[14–16]. The capacity of mESCs to commit into both EPI-primed and PE lineages provides a useful model to study the molecular mechanisms governing these early cell fate decisions[12,13,17–19].

Selective inhibition of the FGF/ERK pathway together with GSK3 inhibition in mESC culture, using the so-called 2i medium, promotes ground-state pluripotency[20,21], in which the PE-like population is absent[12,13,20]. FGF/MAPK signaling has been proposed to be the key regulator of cell identity within the ICM[4]. However, *Fgf4*[−/−] embryos express *Gata6*, which cannot be maintained after E3.25. This suggests that the onset of the PE program is FGF4-independent[22–24]. Moreover, addition of FGF ligands, including FGF1 or FGF2, to mESC culture does not drive PE specification[4,25], suggesting that alternative pathways may be implicated in this process.

Although several Wnt ligands and components are expressed in the preimplantation blastocyst[26–32], a specific role of the Wnt pathway in mammalian preimplantation development has not yet been described. Wnt signaling includes canonical or β-catenin-dependent and non-canonical or β-catenin-independent pathways[33,34]. Repression of Wnt ligand secretion in naive mESCs, which simultaneously inhibits both canonical and non-canonical Wnt cascades, induces EPI-priming[31], yet interestingly, does not impact early lineage specification at preimplantation stage[10]. Exogenous activation of the Wnt/β-Catenin pathway is required for promoting mESC ground-state of self-renewal[35,36] indicating that pluripotent cells are receptive to external signaling modulation. However, the precise role of exogenous Wnt signaling inhibition in lineage specification of mESCs and during pre-implantation development is still unknown.

Upon Wnt ligand binding, unphosphorylated (active) β-Catenin accumulates in the cytoplasm. Subsequently, β-Catenin translocates to the nucleus, where it interacts with the T-cell factor/lymphoid enhancer factor (TCF/LEF) family of TFs to facilitate gene transcription. There are four TCF/LEF (TCF7, LEF1, TCF7L1 and TCF7L2) members in mammals. In mESCs, TCF7L1 is described as a transcriptional repressor, which limits the steady-state levels of the pluripotency network (*Oct4/Sox2/Nanog*) and of their common target genes[37–39]. Alleviation of TCF7L1-mediated repression, following β-Catenin stabilization, is essential for pluripotency maintenance and self-renewal of mESCs[40–42]. Interestingly, *Tcf7l1* deletion does not prevent the generation of primed epiblast stem cells (EpiSCs), even though it delays it[8,39,43], suggesting a minor role for TCF7L1 in EPI-priming. The role of TCF7L1 in specification to other lineages remains unknown.

Here, we provide evidence that inhibition of Wnt signaling enhances PE cell fate specification during preimplantation development as well as in mESC cultures. We demonstrate that forced expression of *Tcf7l1* efficiently drives mESCs towards a PE cell fate. By contrast, deletion of *Tcf7l1* in preimplantation embryos or in naive mESCs, abolishes their ability to differentiate into PE, without compromising epiblast priming. These TCF7L1-dependent effects are mediated by TCF7L1 binding and repression of essential genes for safeguarding naive pluripotency, such as *Nanog*, and *Prdm14*, causing exit from pluripotency. Notably, TCF7L1 also represses genes crucial for formative and EPI-primed programs including *Otx2*, *Lef1*, and *Dnmt3b*, thereby preventing EPI priming and, hence, driving mESCs towards PE. These findings on the function of TCF7L1 during early development enhance our knowledge of the role played by the Wnt signaling pathway, and its

key transcriptional players, the TCF/LEF factors, in regulating early developmental processes and provide further insights in the process of EPI vs. PE lineage segregation.

## Results

### EPI and PE populations show differential Wnt pathway activity in vitro and in embryos

Metastable subpopulations of naive EPI and PE-like cells arise in heterogeneous mESC culture[13,44]. The naive EPI and PE-like populations can be distinguished using platelet endothelial cell adhesion molecule 1 (PECAM1) and platelet-derived growth factor receptor α (PDGFRα) cell surface markers, respectively[13] (Fig. 1a).

To identify the growth factors and signaling pathways involved in the regulation of EPI and PE lineage specification in mESCs we analyzed publicly available RNA-sequencing (RNA-seq) data and evaluated expression differences in genes associated with naive pluripotency and PE between PDGFRα[−]/PECAM1[+] (naive EPI) and PDGFRα[+]/PECAM1[−] (PE-like) cells[13]. As expected, naive EPI and PE-like cells expressed pluripotency- and PE-specific markers respectively (Fig. 1b). Gene ontology (GO) analysis of the differentially expressed genes (DEGs) between naive EPI and PE-like cells[13] highlighted differences in the regulation of the MAPK cascade and apoptosis pathways, in line with previous publications[5,45–47]. Interestingly, GO analysis suggested a potential role of the canonical Wnt pathway in the regulation of EPI/PE lineage segregation (Fig. 1c; Supplementary Data 1). Compared with the PE-like population, naive EPI cells are characterized by a Wnt-ON state, defined by higher expression of well-described Wnt positive regulators (*Fn1, Wnt7b, Lef1, Rspo1* and *Lgr4*) and lower levels of Wnt negative regulatory genes (*Gsk3b* and *Dkk1*) (Fig. 1d). Sorted PE-like population showed lower levels of total and active β-Catenin (Fig. 1e), and lower levels of the transcriptional activators TCF7 and LEF1 (Supplementary Fig. 1a) compared to naive EPI cells, supporting the notion that PE-like cells are characterized by limited Wnt signaling activity.

Previous studies reported that the canonical Wnt pathway is transcriptionally active in the ICM of preimplantation embryos by using an *Axin2* transcriptional reporter[31] or by immunofluorescence for active β-Catenin[29,48]. However, they did not address whether EPI and PE lineages display differential activity levels of canonical Wnt pathway. Using publicly available single cell RNA-seq (scRNA-seq) analysis of two independent E4.5 preimplantation embryo data sets[49,50], we confirmed the differential activity of the Wnt/β-Catenin pathway in EPI and PE lineages (Fig. 1f, Supplementary Fig. 1b−d, and Supplementary Data 2, 3). EPI cells expressed higher levels of canonical Wnt positive regulators (*Fn1, Lef1* and *Lgr4*) compared to PE cells. By contrast, PE cells expressed higher levels of genes involved in Wnt pathway repression (*Gsk3b, Znfr3* and *Dkk1*) (Fig. 1g and Supplementary Fig. 1e). Therefore, we explored the levels of active β-Catenin in EPI (NANOG[+]) and PE (GATA6[+]) cells in preimplantation embryos at E3.5 and E4.5. Three distinct subpopulations of single-positive NANOG[+] (EPI), single-positive GATA6[+] (PE) and double-positive NANOG[+]/GATA6[+] cells were identified in the ICM of freshly isolated E3.5 blastocysts (Fig. 1h, i). By E4.5, the ICM consisted of only two distinct populations: single-positive NANOG[+] and single-positive GATA6[+] cells in both E4.5 freshly isolated and E3.5 + 24H ex vivo cultured embryos (Fig. 1j and Supplementary Fig. 1f). Although active β-Catenin was primarily localized at the cell membrane, in agreement with its function in adherent junctions[51], we also detected diffused cytoplasmatic and nuclear staining for active β-Catenin (Fig. 1h, j and Supplementary Fig.1f), in accordance with previous reports[29]. Quantification of nuclear and total (membrane+intracellular) active β-Catenin confirmed its lower levels in GATA6[+] cells (green arrows) compared to NANOG[+] cells (red arrows) and co-expressing NANOG[+]/GATA6[+] cells (yellow arrows) at E3.5 or E4.5 blastocyst stages (Fig. 1i, k and Supplementary Fig. 1f−h).

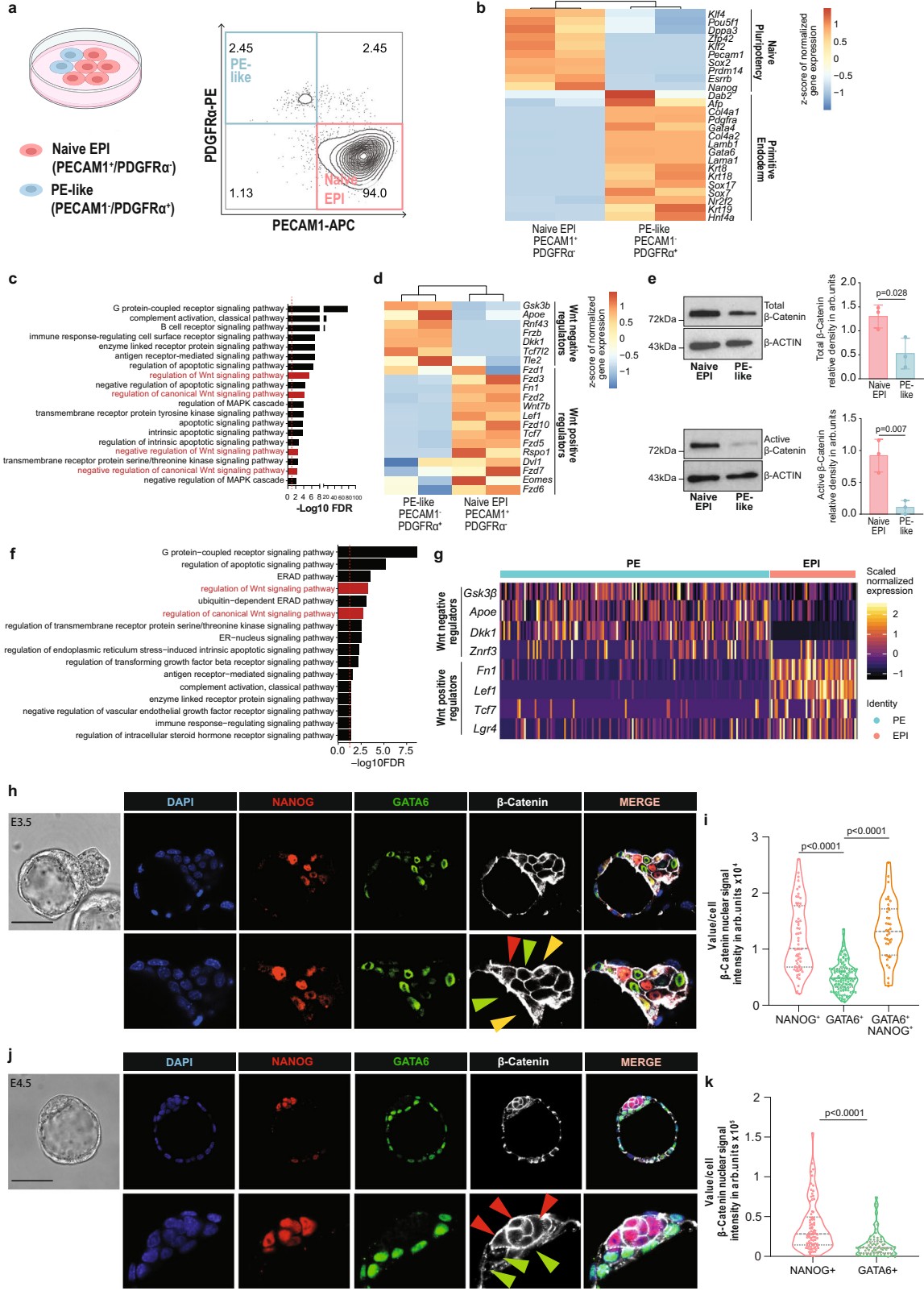

Altogether our findings demonstrate that canonical Wnt-ON and Wnt-OFF states correlate with EPI and PE populations respectively, in heterogeneous mESC cultures and in mouse preimplantation embryos.

## Wnt signaling inhibition promotes a shift towards the PE fate

Previous studies demonstrated an important role of Wnt signaling during post-implantation development or mESC self-renewal[20,21,31,52–56].

Although blocking Wnt ligand secretion in mESC promotes EPI-priming[57], the effect of canonical Wnt inhibition on mouse early lineage specification has not been explored. We therefore assessed if modulation of the canonical Wnt signaling pathway would alter the proportion of naive EPI and PE-like cells in vitro as well as within the ICM of mouse embryos. Treatment of mESCs with the physiological canonical Wnt antagonist Dickkopf1 (DKK1) decreased active β-

**Fig. 1 | EPI and PE populations show differential Wnt pathway activity in vitro and in embryos. a** (Left) Schematic representation of mESCs heterogeneity. Created with BioRender.com. (Right) Flow cytometry density plot of mESC populations co-stained for PECAM1 and PDGFRα. **b** Expression of EPI and PE specific markers by PECAM1[+] and PDGFRα [+] sorted cell populations, respectively. Unsupervised clustering. *n* = 2 independent samples per condition[13]. **c** GO enrichment analysis of DEGs between naive EPI and PE-like sorted cell populations[13]. Red dashed line=FDR 0.05. **d** Expression of Wnt positive and negative regulators in naive EPI and PE-like sorted cell populations. Unsupervised clustering. *n* = 2 independent samples per condition[13]. **e** (Left) Representative images and (right) quantification of total/active β-Catenin levels in naive EPI and PE-like sorted cell populations. Mean ± SD; *n* = 3 biologically independent samples; two-tailed unpaired *t* test. **f** GO enrichment analysis of DEGs between EPI and PE cells in E4.5 embryos from[50]. Red dashed line = FDR 0.05. **g** Expression of Wnt positive and negative regulators in EPI and PE

lineages in E4.5 embryos[50]. **h** Representative immunofluorescence (IF) image of active β-Catenin signal in E3.5 embryo. EPI-NANOG[+] cells (red arrow), PE-GATA6[+] cells (green arrow) and double-positive NANOG[+]/GATA6[+] cells (yellow arrow). Scale bar=50 μM. Zoomed region of interest is reported below the images. **i** Nuclear active β-Catenin signal in EPI-NANOG[+], PE-GATA6[+] and double-positive NANOG[+]/GATA6[+] cells. Integrated intensity in arbitrary units (arb. units). NANOG[+] *n* = 38, GATA6[+] *n* = 111 and NANOG[+]/GATA6[+] *n* = 58 cells; 2 independent experiments; one-way ANOVA test. **j** Representative IF image of active β-Catenin signal in freshly isolated E4.5 embryo. EPI-NANOG[+] cells (red arrow), PE-GATA6[+] cells (green arrow). Scale bar = 50 μM. Zoomed region of interest is reported below the images. **k** Nuclear active β-Catenin signal in EPI-NANOG[+] and PE-GATA6[+] cells. Integrated intensity in arbitrary units (arb. units). NANOG+ *n* = 78; GATA6+ *n* = 68 cells; 2 independent experiments; two-tailed unpaired *t* test. Source data for all experiments are provided as a Source data file.

Catenin protein levels (Supplementary Fig. 2a), causing a significant enrichment in the percentage of PDGFRα[+]/PECAM1[-] (PE-like) cells (Fig. 2a), along with a notable upregulation of PE gene markers, including *Gata6, Gata4,* and *Sox17* (Fig. 2b). This suggested a direct role of Wnt signaling inhibition in PE formation. We also cultured E2.5 embryos for 48 hours to the late blastocyst stage (E2.5 + 48H) in the presence or absence of DKK1. Lineages assessment showed that DKK1-treated embryos contained a significantly increased number of PE (GATA6[+]) cells compared to control embryos. In addition, DKK1-treated embryos showed a strong but statistically non-significant tendency (*p* = 0.055) to have fewer EPI cells (NANOG[+]) than untreated embryos. Interestingly, we did not observe a statistically significant difference in trophectoderm (TE) cell number between control and treated embryos (Fig. 2c, d). Although the total number of ICM cells (Supplementary Fig. 2b) remained unchanged, DKK1-treated embryos exhibited an increased GATA6[+] -PE proportion and a decreased NANOG[+] -EPI proportion (Fig. 2e), demonstrating preferential PE specification at the expense of the EPI upon Wnt inhibition. Interestingly, DKK1-treated embryos showed an accelerated development with more hatched blastocysts at E2.5 + 48H (Supplementary Fig. 2c). A direct correlation between blastocyst lumen expansion and PE specification has been recently shown[58]. Our results show that the blastocyst lumen volume of DKK1-treated embryos was 72% larger (Fig. 2f, g), which was associated with a significant increase of 80% in the embryo size area compared to control embryos (Fig. 2h, i and Supplementary Fig. 2d). Thus, these results demonstrate a direct link between Wnt pathway inhibition and PE specification at the preimplantation stage.

### *Tcf7l1* deletion impairs PE transition without preventing pluripotency exit and neuroectodermal differentiation

To investigate whether PE specification in vitro is mediated by the TCF/LEF transcriptional activity, we used the inhibitor of β-catenin−responsive transcription iCRT3, which prevents the interaction of β-catenin and TCF factors, thus inhibiting transcription of Wnt target genes[59]. iCRT3-treated mESCs exhibited a significantly higher percentage of PDGFRα[+]/PECAM1[-] (PE-like) cells (Fig. 3a). We observed a concomitant enhanced expression of PE markers compared to control (Fig. 3b), indicating that the repressive transcriptional activity of TCF/LEF is directly involved in EPI to PE specification.

TCF7 and TCF7L1 are the most abundant TCF/LEF members in mESCs[60,61]. While TCF7 is considered a Wnt transcriptional activator[62], TCF7L1 is primarily a transcriptional repressor[39,63]. To assess their role in PE formation, we evaluated the effects of *Tcf7* or *Tcf7l1* deletion on mESC heterogeneity. While deletion of *Tcf7l1* resulted in a significant reduction of the endogenous PDGFRα[+]/PECAM1[-] cell population (Fig. 3c, d), the fraction of PE-like cells was not affected by deletion of *Tcf7* (Supplementary Fig. 3a and b).

To further explore the role of TCF7L1 and TCF7 in PE differentiation, WT, *Tcf7l1*[−/−] and *Tcf7*[−/−] mESCs were firstly adapted in 2i medium plus LIF in which the transcriptome of ESCs closely resembles

that of ICM cells[6]. Next, we differentiated the cells using Retinoic Acid (RA) in basal medium, which is known to promote pluripotency exit and induce both extraembryonic PE and embryonic neuroectodermal differentiation[64]. WT cells treated with RA progressively transited towards the PE fate (PDGFRα[+]/PECAM1[-]) yielding more than 50% PE cells by 4 days (Fig. 3e). Strikingly, although *Tcf7l1*[−/−] cells cultured with RA lost expression of the PECAM1 pluripotency marker, they did not differentiate into PE-like cells (Fig. 3e, f) and PE marker gene expression was not induced in RA-treated *Tcf7l1*[−/−] cells (Fig. 3g). Furthermore, the percentage of GATA6[+] cells was considerably lower in *Tcf7l1*[−/−] cells treated with RA (Fig. 3h, i). On the contrary, *Tcf7*[−/−] cells were able to undergo PE specification after RA treatment, as shown by the high expression of PE markers, which was accompanied by pluripotency exit (Supplementary Fig. 3c−e).

TCF7L1 binds and represses pluripotency markers, suggesting that the *Tcf7l1*-dependent PE differentiation defect might be caused by impaired pluripotency exit[39,62]. Although decrease of pluripotency gene expression was delayed in *Tcf7l1*[−/−] cells at early differentiation time points (D1 and D2) as previously reported[8], *Tcf7l1*[−/−] cells were able to efficiently exit pluripotency after D3 of RA treatment similarly to WT (Fig. 3j, k). To evaluate whether *Tcf7l1*-dependent differentiation impairment was related to other lineages we investigated the expression of neuroectodermal markers, which was also induced by RA treatment. Interestingly, both WT and *Tcf7l1*[−/−] mESCs exhibited strong and similar upregulation of neuroectodermal genes and showed comparable percentages of NESTIN[+] cells at D4 of RA treatment[65] (Fig. 3k, l).

In summary, we demonstrated that *Tcf7l1* deletion in mESCs delays but does not abrogate pluripotency exit. Furthermore, *Tcf7l1*[−/−] cells increase neuroectodermal marker expression to the same level as WT cells upon RA treatment, suggesting that *Tcf7l1*[−/−] cells maintain their embryonic differentiation potential. In contrast, *Tcf7l1*[−/−] mESCs are incapable of converting to PE-like cells, illustrating an essential role of TCF7L1 in extraembryonic PE induction in vitro.

### *Tcf7l1* expression is sufficient to drive PE cell fate specification

Forced expression of PE-lineage specific TFs[14–16] or repression of naive EPI-specific TFs[66,67] promotes mESC differentiation towards PE. However, the putative role of non lineage-specific TFs in PE formation has yet to be studied. To examine the role of TCF7L1 in PE cell fate specification in mESCs, we performed overexpression analysis using a doxycycline (Dox)-inducible (Tet-OFF) *Tcf7l1* ESC line[68]. Following Dox removal for 24H, we observed robust expression of the TCF7L1-FLAG tag protein along with Venus YFP (Supplementary Fig. 4a, b). However, not all cells responded equally to the Dox removal as it can be seen from YFP fluorescence intensity variations (Supplementary Fig. 4b), possibly due to time and cellular variations of response to Dox removal[69]. We confirmed that *Tcf7l1* overexpression correlated with Venus[high] expression (Supplementary Fig. 4c). To study the effect of uniform overexpression (OE) of *Tcf7l1* in mESC culture, we sorted Venus[high] cells (Supplementary Fig. 4c) 24H after Dox removal (D1) and

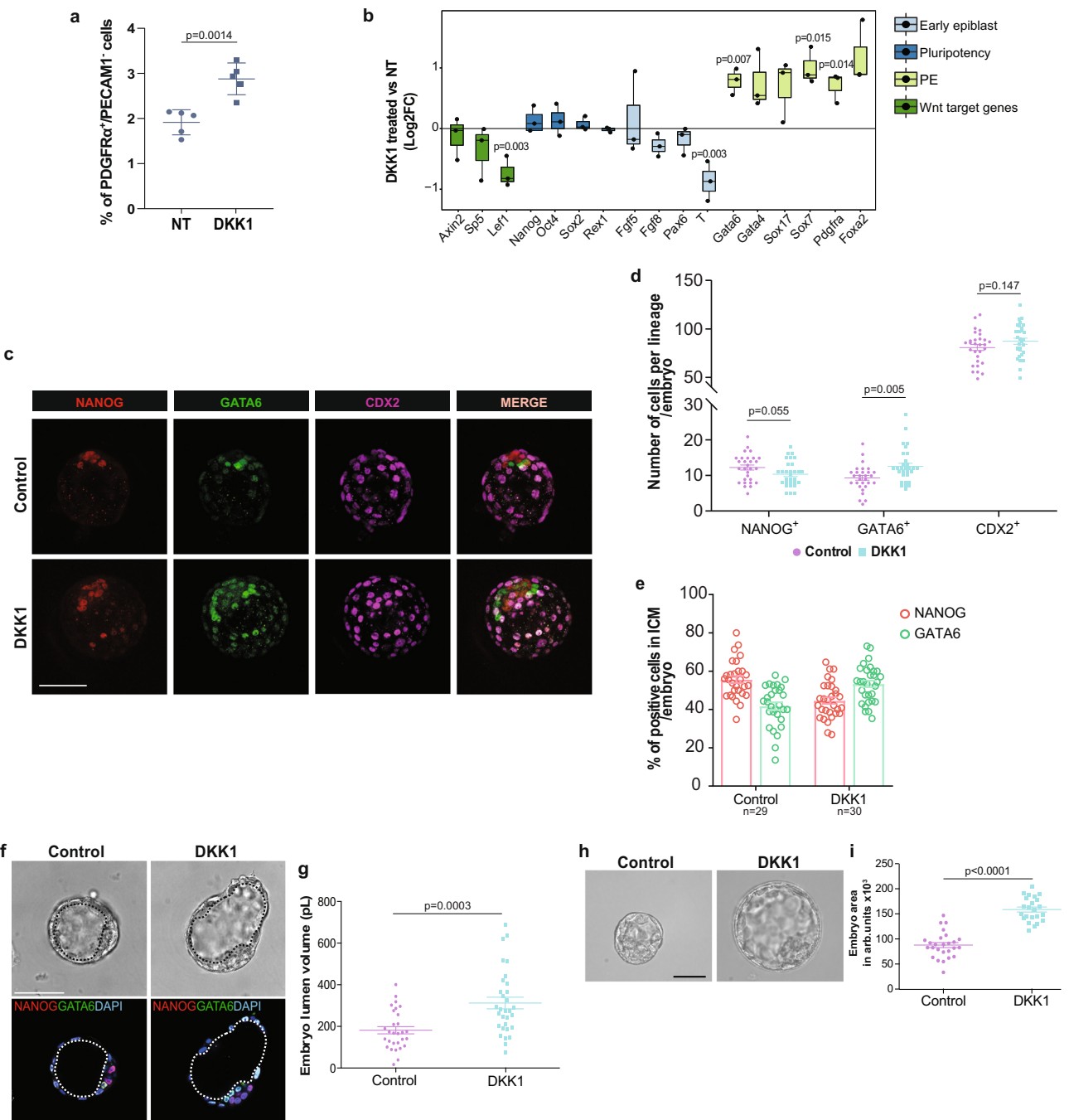

**Fig. 2 | Wnt signaling inhibition promotes a shift towards the PE fate.**
**a** Percentage of PE-like cells (PDGFRα⁺/PECAM1⁻) in control and DKK1-treated mESCs. Mean ± SD; $n$ = 5 independent experiments, two-tailed unpaired $t$ test. **b** Gene expression analysis of early EPI, pluripotency, PE and Wnt target genes on control and DKK1-treated mESCs. Horizontal line denotes the median value, box refers to the 25th to 75th percentiles and whiskers mark min and max values. $n$ = 3 independent experiments; two-tailed unpaired $t$ test. **c** Representative IF image of NANOG, GATA6 and CDX2 protein signals in E2.5 + 48H control and DKK1 treated embryo. Scale bar = 50 μM. **d** Number of NANOG⁺,GATA6⁺ and CDX2⁺ cells (counts) per embryo. Mean ± SEM; Control: $n$ = 29; DKK1 n = 30; 3 independent experiments;

multiple unpaired $t$ tests with Holm-Sidak method. **e** Percentage of NANOG⁺ and GATA6⁺ cells normalized on total number of ICM per embryo. Mean ± SEM; Ctrl $n$ = 29, DKK1 $n$ = 30; 3 independent experiments. **f** Representative BF and IF image of E2.5 + 48H control and DKK1 treated embryo. Black and white dotted lines delimitate blastocyst cavity (lumen). Scale bar = 50 μM. **g** Embryo lumen volume reported in pico liters (pL). Mean ± SEM; Ctrl $n$ = 29, DKK1 $n$ = 31; 3 independent experiments. two-tailed unpaired $t$ test. **h** Representative brightfield (BF) image of embryo morphology. Scale bar = 50 μM. **i** Embryo total area reported in arb. units Mean ± SEM; Ctrl $n$ = 26, DKK1 $n$ = 23; 3 independent experiments, two-tailed unpaired $t$ test. Source data for all experiments are provided as a Source data file.

replated the cells for an additional 7 days under ESC self-renewal conditions. By day 4, Venus^high cells exhibited a flat morphology reaching an epithelial-like morphology by day 8 (Fig. 4a). After 48H of *Tcf7l1* induction, we observed progressive upregulation of the PE genes *Gata6, Gata4, Pdgfrα,* and *Sox17*, reaching levels comparable to

those of embryo-derived extraembryonic endoderm stem (XEN) cells by day 8 (Fig. 4b), along with decreased expression of the pluripotency genes *Nanog* and *Sox2* (Fig. 4c). Furthermore, 6 days of *Tcf7l1* induction resulted in an increased number of GATA6⁺ cells, which was accompanied with decreased NANOG⁺ cells (Fig. 4d, e and

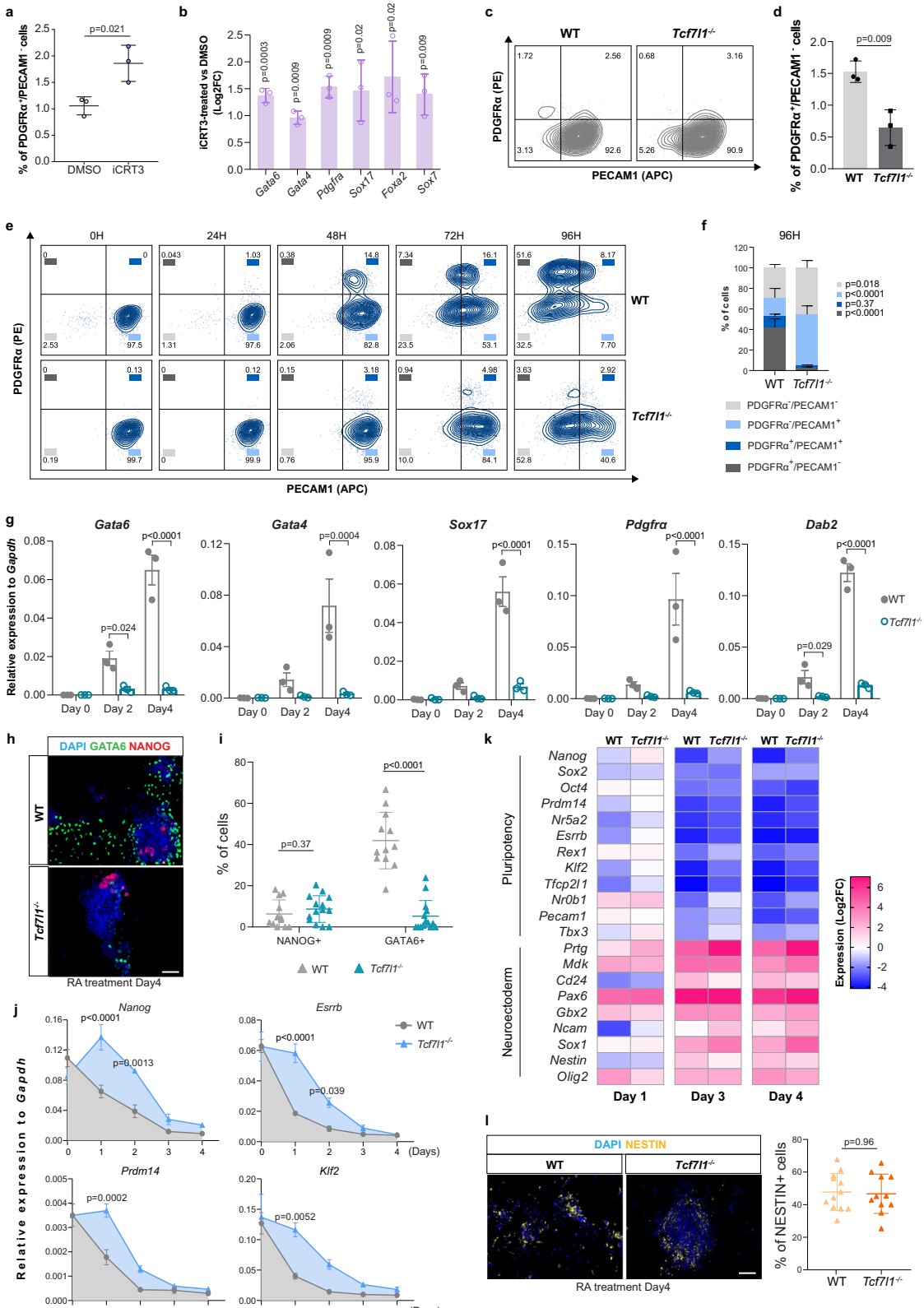

(Supplementary Fig. 4d). By contrast, forced expression of *Tcf7* (Supplementary Fig. 4e, f), resulted in negligible changes in PE gene expression (Supplementary Fig. 4g), indicating that TCF7 has a minor role, if any, in PE lineage specification.

Next, we performed RNA-seq of D0 and D8 *Tcf7l1*-OE cells to assess gene expression patterns resulting from *Tcf7l1* overexpression. We identified 2063 DEGs between D0 and D8 (|log2FC| >2, FDR ≤ 0.05)

(Supplementary Data 4). GO analysis of DEGs showed association with embryo morphogenesis and endoderm differentiation processes (Supplementary Fig. 4h and Supplementary Data 4). In addition, gene expression profile of D8 *Tcf7l1*-OE cells showed that, unlike Wnt secretion inhibition which promotes EPI-priming[31], overexpression of *Tcf7l1* reduced primed and naive pluripotency related gene expression (Fig. 4f). Visceral endoderm (VE) and Parietal endoderm (ParE) are major

**Fig. 3 | *Tcf7l1* deletion impairs PE transition without preventing pluripotency exit and neuroectodermal differentiation. a** Percentage of PDGFRα⁺/PECAM⁻ cells in mESC cultured with iCRT3. Mean ± SD; *n* = 3 independent experiments; two-tailed unpaired *t* test. **b** qRT-PCR of PE markers in mESCs cultured with iCRT3. Values represent Log₂ of fold change expression relative to DMSO. Log₂FC ± SD; *n* = 3 independent experiments; multiple unpaired *t* tests with Holm-Sidak method. **c** Representative flow cytometry plots of WT and *Tcf7l1⁻/⁻* cells co-stained for PECAM1 and PDGFRα. **d** Percentage of PDGFRα⁺/PECAM⁻ cells in WT and *Tcf7l1⁻/⁻* mESCs. Mean ± SD; *n* = 3 biologically independent samples; two-tailed unpaired *t* test. **e** Representative flow cytometric plots of WT and *Tcf7l1⁻/⁻* cells co-stained for PECAM1 and PDGFRα markers upon RA treatment. *n* = 3 independent experiments. **f** Flow cytometry analysis of PDGFRα and PECAM1 populations at 96H of RA treatment. Mean ± SD; *n* = 3 independent experiments; two-way ANOVA test. **g** qRT-PCR of PE markers in WT and *Tcf7l1⁻/⁻* mESCs upon RA treatment. Mean ± SEM; *n* = 3;

two-way ANOVA test. **h** Representative IF image of GATA6 and NANOG in WT and *Tcf7l1⁻/⁻* mESCs upon RA treatment. Scale bar = 50 μm. **i** Quantification of NANOG⁺ and GATA6⁺ cells of Fig. 3h. Mean ± SD; two-tailed unpaired *t* test. Each dot represents % of cells in a field of view; WT:*n* = 12 non-overlapping images, *Tcf7l1⁻/⁻*: *n* = 14 non-overlapping images from *n* = 1 experiment. **j** qRT-PCR of pluripotency markers in WT and *Tcf7l1⁻/⁻* mESCs upon RA treatment. Mean ± SEM; *n* = 3 independent experiments; two-way ANOVA test. **k** qRT-PCR of embryonic neuroecto-dermal markers in WT and *Tcf7l1⁻/⁻* mESCs upon RA treatment. Gene expression values are reported as Log₂ of fold change expression. *n* = 3 independent experiments. **l** (Left) Representative IF image of NESTIN in WT and *Tcf7l1⁻/⁻* mESCs upon RA treatment. (Right) Quantification of NESTIN⁺ cells. Mean ± SD; two-tailed unpaired *t* test. Each dot represents % of cells in a field of view; WT: *n* = 12 non-overlapping images, *Tcf7l1⁻/⁻*: *n* = 11 non-overlapping images from *n* = 1 experiment. Source data for all experiments are provided as a Source data file.

derivatives of primitive endoderm and are distinguished by characterized markers[70–74]. Interestingly, while XEN cells transcriptionally resemble more ParE[71,72], *Tcf7l1*-OE-mESCs displayed high levels of PE and VE markers expression suggesting that OE of *Tcf7l1* promotes a VE- rather than a ParE-cell fate (Fig. 4g). Overall, these findings demonstrate that upon forced expression of *Tcf7l1*, cells engage in PE gene activation, revealing the key role played by TCF7L1 in PE differentiation of mESCs. Whether these cells can contribute to the extraembryonic layers of mouse embryos still needs to be examined.

## PE cell fate specification is preceded by naive and formative pluripotency repression by TCF7L1

To define the transcriptional dynamics regulating early embryo development and PE formation, we performed transcriptomic analysis of *Tcf7l1*-OE-mESCs at different timepoints. *Tcf7l1*-OE-mESCs were collected 24H, 48H and 96H after Dox removal along with Dox-treated cells as control for RNA-seq analysis. Gene expression profiling identified 856 DEGs on D1, 1663 DEGs on D2 and 1656 DEGs on D4 between *Tcf7l1*-OE and Dox-treated mESCs (Supplementary Data 5). GO analysis of downregulated genes after 1 day of *Tcf7l1*-OE showed enrichment of pathways involved in the regulation of embryonic development, cell fate specification and negative regulation of cell adhesion along with the known developmental pathways, BMP and FGF (Fig. 5a). In addition, GO terms and KEGG pathways related to stem cell development and differentiation together with terms associated with canonical Wnt pathway were identified in D1 *Tcf7l1*-OE-mESCs (Fig. 5a and Supplementary Data 6). Specifically, Wnt target genes (*Axin2, Lef1 and* Sp5) as well as genes associated with naive and general pluripotency (*Nanog, Klf2, Prdm14, Dpp4* and *Fn1*) were downregulated. In line with previous reports[75], we found little effect on *Oct4* expression on D1, demonstrating that PE precursors retain *Oct4* expression at early stages[13,76].

Recently, formative pluripotency has been described as an intermediate phase that precedes primed epiblast formation upon naive pluripotency exit[8,9,11]. The formative state is characterized by decreased expression levels of naive pluripotency markers and increased expression of peri-implantation markers such as *Lef1* and *Dmnt3a* and *Dnmt3b*. Expression of *Otx2* has also been shown to be essential for the maintenance of the formative pluripotency state[8,77,78]. Overexpression of *Tcf7l1* caused a decrease in expression of several formative-specific genes including *Otx2, Lef1, Dnmt3b, Fgf5, Pim2* and *Zic2*[79] already on D1, which was maintained on D2 and D4 (Fig. 5b–d).

We did not observe increased expression of endoderm genes on D1 (Fig. 5b). However, on D2, endodermal genes were induced, which was even more obvious 4 days after OE of *Tcf7l1* (Fig. 5c, d). Upregulated genes on D4 were associated with the formation of endoderm, cell fate determination, the PDGFRα signaling pathway as well as gastrulation and anterior/posterior pattern specification (Fig. 5e and Supplementary Data 7). We observed increased levels of primitive, parietal and visceral endoderm genes, including master regulators of endoderm formation (*Gata6, Gata4* and *Sox17*), genes involved in cell

adhesion (*Col4a1, Col4a2* and *Dab2*) and other genes associated with PE function (*Pdgfra, Sparc, Thbd, Nid1* and *Cited1*) (Fig. 5f). These findings demonstrate that activation of PE genes follows the downregulation of naive and formative pluripotency.

Using a TF-gene target enrichment analysis[80,81], we found that the genes downregulated after *Tcf7l1* OE were putative targets of TCF7L1, CTNNB1 (β-Catenin), NANOG, SOX2 and OCT4 (Supplementary Fig. 5a), confirming the coregulation of TCF7L1 targets with the pluripotency TF network[37].

To further assess whether TCF7L1 directly regulates formative pluripotency gene expression, we integrated RNA-seq data of genes downregulated on D1 in *Tcf7l1*-OE-mESCs with publicly available TCF7L1 ChIP-seq data on mESCs[36]. This revealed that 55 genes (*p* = 0.0021) that were downregulated on D1, were also bound by TCF7L1 around the TSS (transcription start site) region (Fig. 5g, Supplementary Fig. 5b and Supplementary Data 8). Common genes included naive (*Nanog, Tdh, Tbx3, Klf2* and *Prdm14*), and formative (*Dnmt3b, Otx2, Sox12* and *Lef1*) pluripotency regulators. Interestingly, bound genes were associated with stem cell population maintenance, blastocyst formation and embryonic development together with signaling pathways regulating pluripotency and canonical Wnt signaling (Fig. 5h and Supplementary Data 9).

Hence, our results demonstrate direct repression of naive and formative pluripotency genes by TCF7L1, which might prevent formative- and consequently EPI-priming. To test this hypothesis, we performed RT-qPCR gene expression analysis of WT, *Tcf7l1⁻/⁻* and *Tcf7l1*-OE mESCs during epiblast-like cell (EpiLCs) differentiation[9]. WT and *Tcf7l1⁻/⁻* mESCs successfully downregulated naive and general pluripotency markers and upregulated key genes involved in formative and primed EPI differentiation[8] (Fig. 5i). *Tcf7l1*-OE cells showed comparable downregulation of naive and general pluripotency genes, yet a decreased upregulation of formative genes compared to WT and *Tcf7l1⁻/⁻* cells (Fig. 5i). Interestingly, *Tcf7l1*-OE cells displayed increased expression of PE genes in EpiLCs differentiation conditions, indicating that forced expression of *Tcf7l1* has a dominant effect, promoting PE differentiation even in non-permissive PE cell culture conditions (Fig. 5i). These results confirm that TCF7L1 restrains formative and subsequent primed transition, enforcing cells to PE specification.

## *Tcf7l1* is required for PE lineage formation in embryos

Using publicly available scRNA-seq and single-cell gene regulatory inference and clustering (pySCENIC) data from mouse embryos[82], we assessed the activity of TCF/LEF TFs along with their target genes (known as regulons) at the preimplantation stage. Interestingly, regulon analysis predicted that Tcf7- and Lef1-regulons were transcriptionally active primarily in the EPI compartment, whereas Tcf7l1- and Tcf7l2-regulons appeared transcriptionally active in PE cells (Fig. 6a, b). To unveil the role of TCF/LEF factors during PE specification, we used published scRNA-seq data, where the Harmony algorithm combined

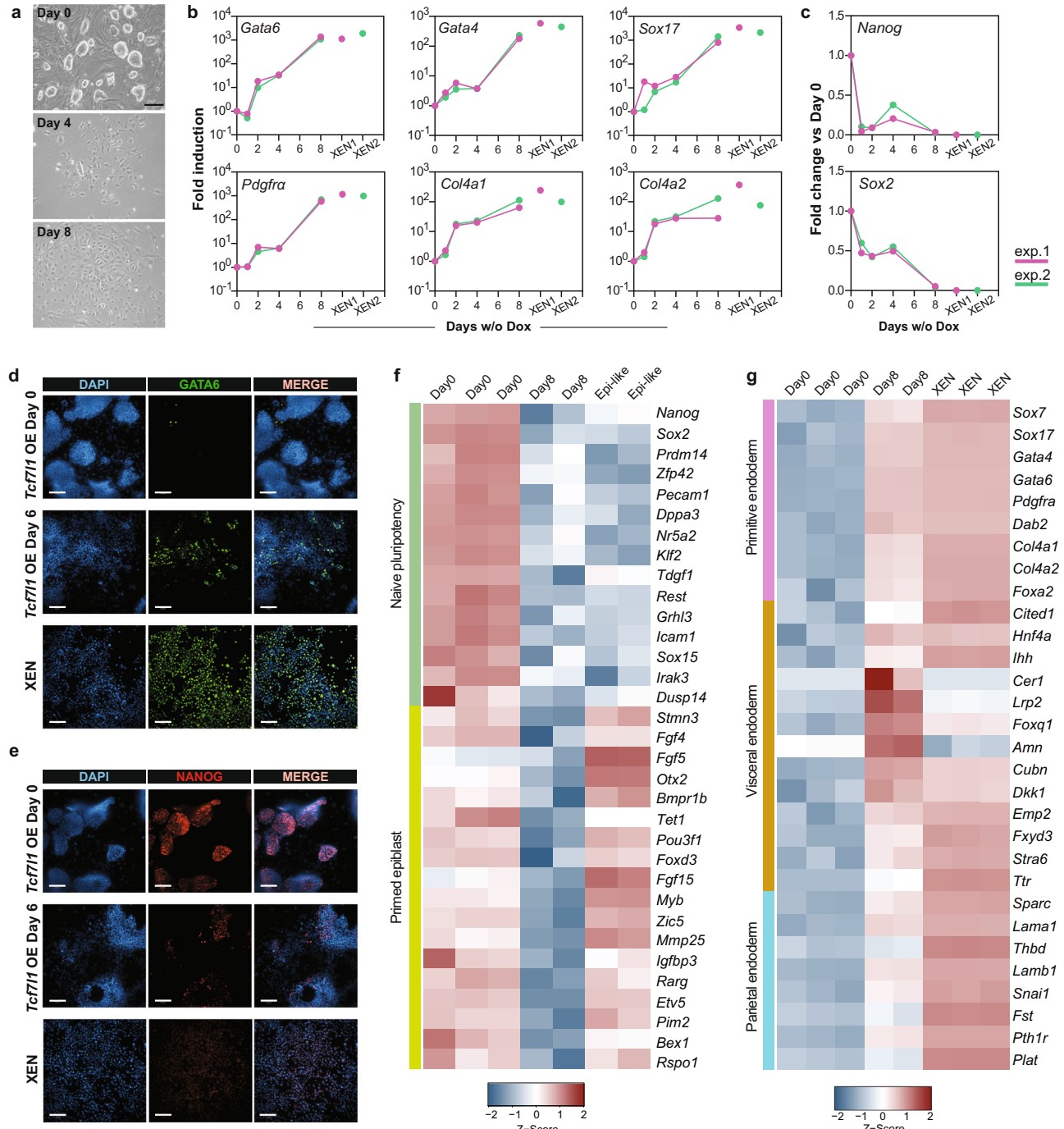

**Fig. 4 | *Tcf7l1* expression is sufficient to drive PE cell fate specification.**
**a** Representative BF images of *Tcf7l1*-OE-mESCs cultured with Dox (D0) or without Dox (D4 and D8) from 3 independent experiments. Scale bar = 50 μm. **b** qRT-PCR of PE markers at indicated time points of *Tcf7l1* OE from 2 independent experiments (pink and green lines). Expression levels of each gene is shown for XEN cells from the corresponding experiment (pink and green dots). All values are reported as fold change relative to Day0. The *x*-axis reports log₁₀ scale. $n = 2$ independent experiments, see Source data file. **c** qRT-PCR of pluripotency markers at indicated time points of *Tcf7l1* OE from 2 independent experiments (pink and green lines). Expression levels of each gene is shown for XEN cells from the corresponding

experiment (pink and green dots). All values are reported as fold change relative to Day0. $n = 2$ independent experiments, see Source data file. **d** Representative IF image of GATA6 in *Tcf7l1*-OE-mESCs after 0 and 6 days of induction and XEN cells from 3 independent experiments. Scale bar = 100 μM. **e** Representative IF of NANOG as in Fig. 4d. **f** Normalized gene counts heatmap showing expression of naive pluripotency and primed epiblast markers in mESCs (D0), after 8 days of *Tcf7l1* induction (D8) and in Epi-like cells. **g** Normalized gene counts heatmap showing expression of primitive endoderm, visceral and parietal endoderm markers in mESCs (D0), after 8 days of *Tcf7l1* induction (D8) and XEN cells. Source data are provided in the Source data file.

with Palantir were used to construct a spatio-temporal map of PE specification in mouse embryos[50]. *Tcf7l1* and *Tcf7l2* showed distinct expression patterns during preimplantation development. Specifically, we found a transient upregulation (pulse) of *Tcf7l1* in PE-fated cells between E3.5 and E4.5, which is compatible with the involvement of

TCF7L1 in PE formation (Fig. 6c). In line with our previous data on mESCs, naive and formative pluripotency markers were significantly decreased, whereas PE genes were considerably induced, following the *Tcf7l1* pulse (Fig. 6c). By contrast, EPI-fated cells lacked the *Tcf7l1* pulse in gene expression levels (Supplementary Fig. 6a). To further elucidate

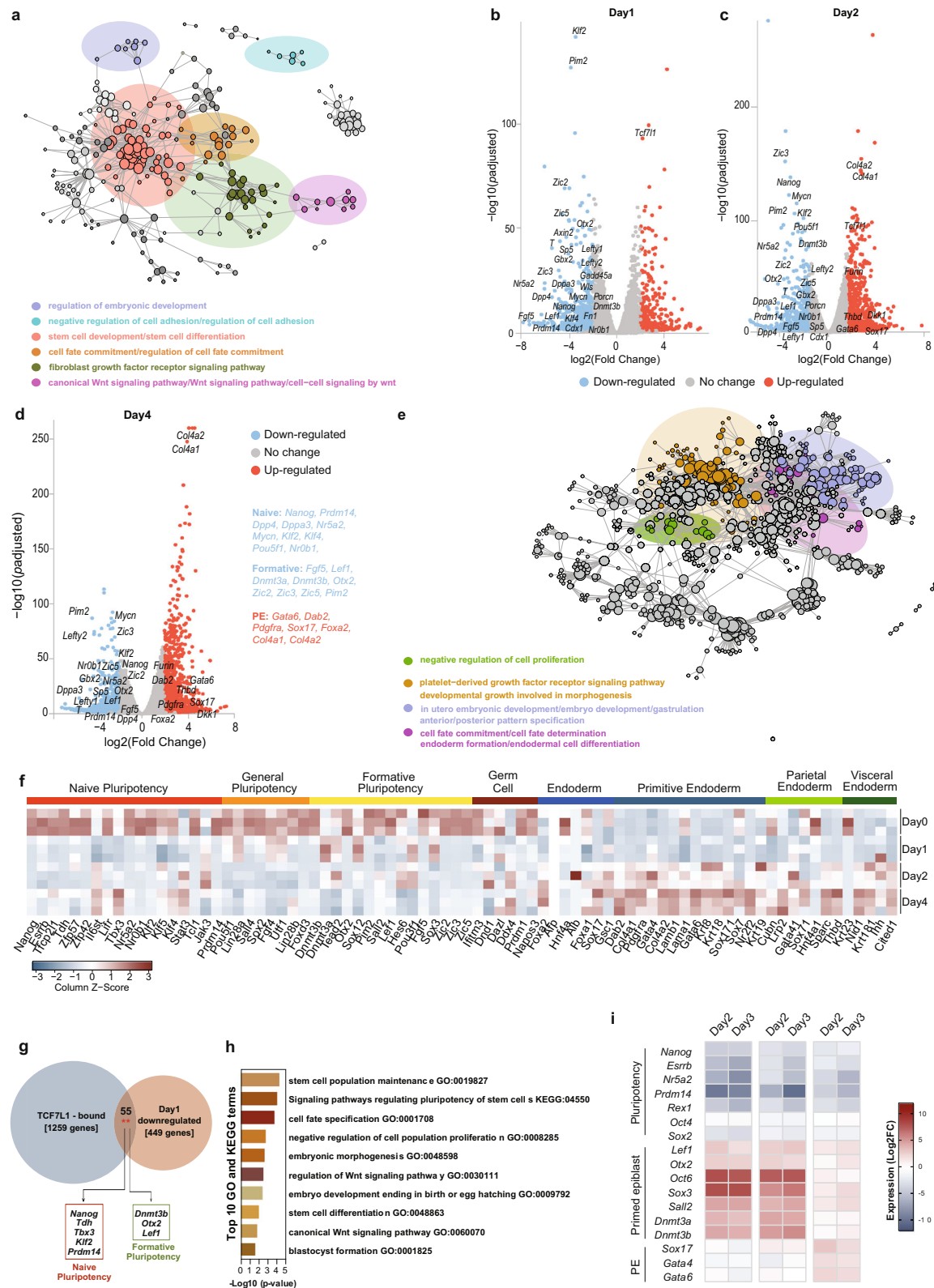

the role of TCF/LEF factors during development, E2.5 embryos were treated ex vivo with iCRT3 for 48H (Fig. 6d). This resulted in embryos with a significantly increased number of GATA6⁺ cells, fewer NANOG⁺ cells (Fig. 6e, f) and a larger lumen volume (Fig. 6g), phenocopying the effects exerted by DKK1 (Fig. 2). This suggests that TCF/LEF factors play an essential role in regulating developmental processes such as cavitation and PE cell fate specification.

Next, we used CRISPR/Cas9-mediated genome editing to further delineate the role of TCF7L1 in regulating cell lineage specification during preimplantation development (Fig. 6h). *Tcf7l1*-targeting ribonucleoproteins (RNPs) were microinjected at the zygote stage (E0.5), resulting in complete editing of the *Tcf7l1* locus, caused mainly by out-of-frame indel mutations (Supplementary Data 10). Only embryos with a 100% efficiency on *Tcf7l1* genome editing were considered for

**Fig. 5 | PE cell fate specification is preceded by naive and formative pluripotency repression by TCF7L1. a** GO and KEGG functional enrichment analysis for *Tcf7l1*-OE-mESCs after 24 H of induction. Functional term networks were obtained using Cytoscape's ClueGO plugin. Each node corresponds to an enriched term and edges connect terms that share a significant number of genes. Network communities (shown in color) were created using the Louvain algorithm, adjusted for betweenness centrality. **b**–**d** Volcano plots showing down- (blue dots) and upregulated (red dots) genes altered on different timepoints of *Tcf7l1* induction. Gene values are reported as a Log$_2$FoldChange. Differential expression analysis was performed with a two-tailed Wald's test. All *p*-values were adjusted for multiple comparisons using Benjamini-Hochberg correction. Annotated points correspond to selected marker genes. **b** DEGs after 24H of induction. **c** DEGs after 2 days of induction. **d** DEGs after 4 days of induction. **e** GO and KEGG functional enrichment analysis in the *Tcf7l1*-CRISPR/Cas9 group (Supplementary Fig. 6b, c). analysis for *Tcf7l1* mESCs after 4 days of induction. Network was created and analyzed as described in Fig. 5a. **f** Normalized gene counts heatmap showing 80 selected specific lineage markers in WT and *Tcf7l1*$^{-/-}$ cells upon 1, 2 and 4 days of induction. **g** Venn diagram indicating genes bound by TCF7L1 (1259 genes) from publicly available ChIP-seq data[36] and downregulated genes in *Tcf7l1*-OE-mESCs after 1 day of induction (449 genes with unique promoters). There were 55 genes shared between the two sets (p-value of a two-tailed Fisher's exact test = 0.0021) which include markers of naive and formative pluripotency. **h** Top 10 functional enrichment terms for the 55 genes downregulated in *Tcf7l1*-OE-mESCs after 1 day of induction and bound by TCF7L1. **i** qRT-PCR of pluripotency, primed epiblast/formative, and PE gene markers in WT, *Tcf7l1*$^{-/-}$ and *Tcf7l1*-OE-mESCs upon EpiLCs differentiation. Gene expression values are reported as Log$_2$ of fold change expression, see Source data file.

Control groups consisted of negative control RNP-injected zygotes and WT control zygotes. No difference in the total number of ICM cells was seen between *Tcf7l1*$^{-/-}$ embryos and controls (Fig. 6i). However, we observed a profound shift in PE towards EPI cell fate specification in *Tcf7l1*$^{-/-}$ embryo ICMs, that now contained a significantly greater number of NANOG$^+$ (EPI) cells and a reduced number of GATA4$^+$ (PE) (Fig. 6j). Also, the *Tcf7l1*$^{-/-}$ blastocyst lumen volume was substantially smaller compared to control embryos (Fig. 6k).

Altogether this study reveals a previously undescribed TCF7L1-dependency in PE formation during preimplantation development. Absence of TCF7L1 induces accumulation of EPI (NANOG$^+$) cells with a significant reduction of PE (GATA4$^+$) cells.

## Discussion

In this study we investigated the role of canonical Wnt inhibition in preimplantation development and mESC fate in vitro. We showed that extrinsic inhibition of Wnt by DKK1 treatment or by transcriptional inhibition (iCRT3) drives PE formation during preimplantation development and mESC cell fate decisions. Furthermore, forced expression of the Wnt-transcriptional repressor, *Tcf7l1*, is sufficient to induce differentiation of mESCs into PE-like cells, but not into EPI-derived lineages. Conversely, ablation of *Tcf7l1* expression diminishes PE formation during preimplantation development and impairs PE differentiation in mESCs without compromising EPI-priming.

The co-emergence of EPI and PE lineages within the ICM of the preimplantation blastocyst requires initial co-expression of the lineage-specific markers NANOG and GATA6[4]. The ability of NANOG and GATA6 to antagonize each other[14,17,83] suggests that either activation or repression of any of the respective lineage-transcriptional programs may provide the basis for the binary EPI or PE cell fate specification. However, the initial developmental cues that drive the expression of different transcriptional regulators, linked with the acquisition of EPI or PE fate in vivo, as well as the PE lineage conversion in mESCs, remain unclear.

This work already highlights TCF7L1 as part of the initial heterogeneity in the ICM and as a potential candidate for driving PE specification. Notably, TCF7L1 is an intrinsically expressed TF in naive pluripotent mESCs[4,61] and in the preimplantation embryo. Pseudotime trajectory generated from published scRNA-seq data on mouse embryos revealed progressive upregulation of PE-specific genes following the "pulse" of *Tcf7l1* expression. This agrees with our RNA-seq data on mESCs where PE gene transcription was induced upon *Tcf7l1* overexpression, underlying the similarity with the in vivo expression patterns and highlighting the role of TCF7L1 as a rheostat and a crucial component of binary EPI vs. PE lineage decisions. The integrated time-series RNA-seq and ChIP-seq data presented here revealed that TCF7L1 binds and represses genes related to formative transition, a prerequisite state prior to primed EPI specification, while we did not identify binding of TCF7L1 on PE promoters. In fact, upon transcriptional Wnt inhibition by *Tcf7l1* overexpression, TCF7L1 dominates and represses embryonic lineage-determining TFs, allowing the PE program to take over and to initiate PE specification.

mESCs resemble the embryonic EPI lineage and are strongly biased towards embryonic fates. This is also reflected by the accumulation of repressive chromatin marks on a subset of promoters, including the PE-specific *Gata6* promoter[6,84]. Nonetheless, previous studies have shown that mESCs also have the potential to transit spontaneously towards an extraembryonic PE-like state[12–16]. Naive pluripotency is supported by the combined modulation of two cascades, the activation of Wnt via GSK3 inhibition, and the inhibition of Erk[21]. Interestingly, MAPK/Erk activation via FGF4 treatment of mouse blastocysts has been reported to induce pan-ICM PE fate[4,22,23]. Several studies have corroborated the role of FGF/MAPK signaling in the resolution of the co-expression state and the EPI vs. PE cell fate establishment but also in the maintenance of *Gata6* expression in the PE population[22–24]. The prevailing model until recently posited the reciprocal expression of *Fgf4* and *Fgfr2* in EPI-biased and PE-biased cells respectively as the driving force of PE fate establishment[23,85]. The latest model highlights the role of an additional FGF receptor, *Fgfr1*, which is expressed in all ICM cells at earlier stage than *Fgfr2* and can equally transduce the FGF4 signal[86]. Transduction of this signal through FGFR1 stimulates low Erk activity and facilitates the acquisition of the EPI fate by a subset of cells. Upon *Fgfr2* expression by the presumptive PE cells, robust Erk activation through both FGFR1 and FGFR2 leads to complete suppression of the EPI program, thereby activating the PE program. However, it has been proposed that this network of regulation is not solely centered on the repressive input of FGF/MAPK signaling on the EPI program but also involves an additional positive input of it onto GATA6 expression, directly activating the PE program[87]. Equivalent to embryos, in vitro experiments have demonstrated that Erk activation induces PE priming in naive mESCs[88]. Our findings support that, similarly to MAPK/Erk activation[88], Wnt inhibition promotes extraembryonic PE differentiation but not differentiation towards embryonic epiblast-derived lineages. Altogether, these results insinuate that maintenance of ground-state pluripotency conditions by both Erk and Wnt signaling modulation relies on the suppression of PE differentiation rather than the active induction of a naive pluripotent state.

Although it is well known that the canonical Wnt pathway is indispensable for peri- and post-implantation development[48], its role in preimplantation stages remains unclear. Expression of a stabilized isoform of β-catenin[89], deletion of *β-catenin*[90], *Apc* loss-of-function[91] or deletion of *Tcf7l1*[92], all affect development progression at post-implantation stages in mouse embryos; however, effects on EPI vs. PE specification were not studied in these cases. It has been shown that Wnt agonists, antagonists and components involved in the transduction machinery are expressed in the mouse preimplantation embryo including *Wnt1*, *Wnt3a*, *Srfp1* and *Dkk1*[28,93]. Interestingly, *Porcn*$^{-/-}$ embryos, which are defective for Wnt secretion, did not show any defect on cell number or cell fate decisions in preimplantation development, indicating that autocrine embryo Wnt signaling activation is

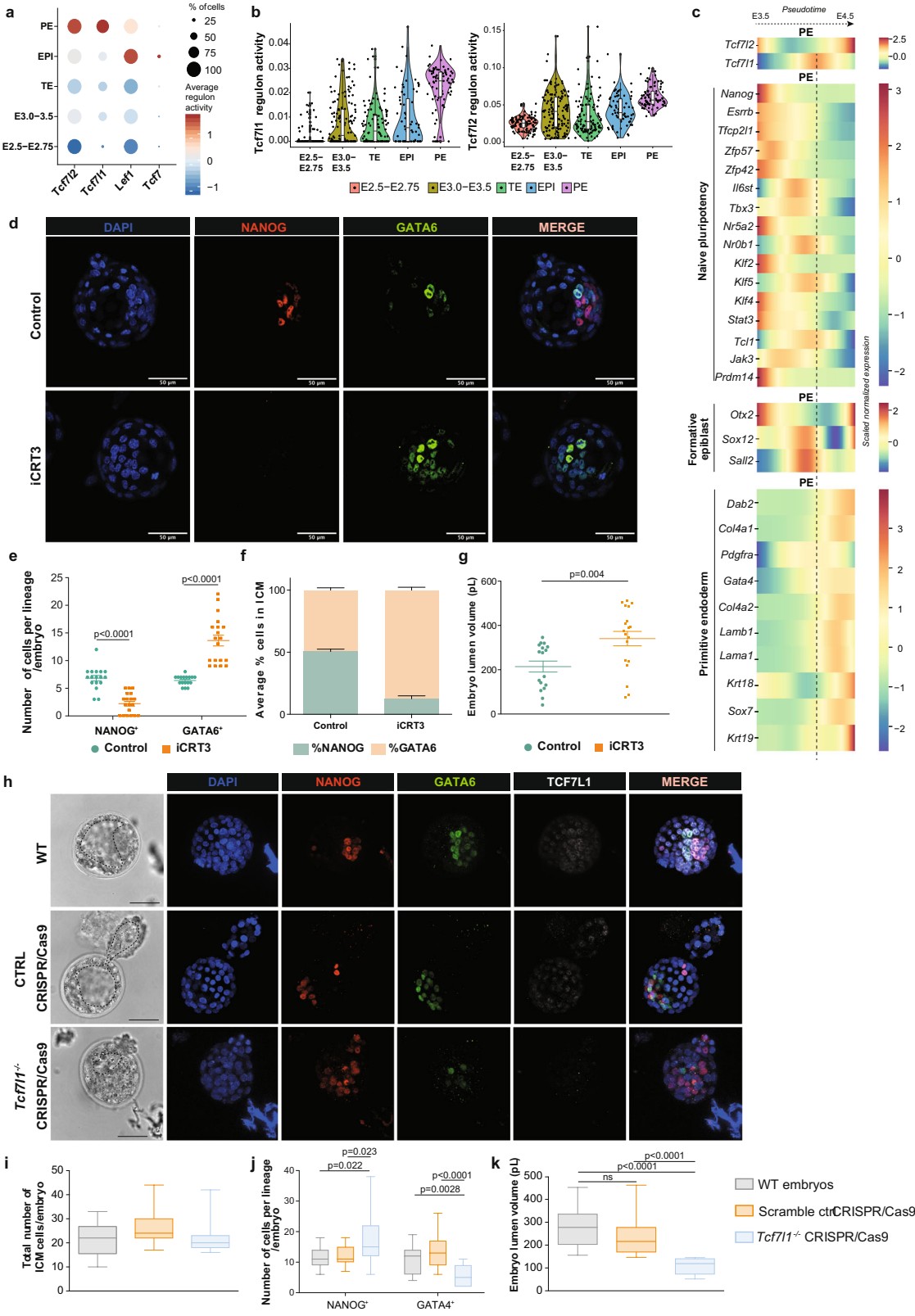

dispensable at these stages. In accordance, ligands and inhibitors of Wnt signaling are expressed in the murine and/or human endometrium in unique patterns at different stages of the menstrual cycle, indicative that embryo development does not solely rely on autologous Wnt signals[52,94,95]. In support of this, our results show that extrinsic inhibition of the Wnt pathway leads in a significant increase of PE cells and reduced EPI cell number, suggesting that the Wnt pathway

is important for EPI and PE lineage segregation. We speculate that the post-implantation defects caused by Wnt component deletion might originate from the defective EPI vs. PE segregation during pre-implantation development.

Based on in vitro experiments, it was previously considered that ablation of *β-catenin* had little or no effect on the self-renewal and transcriptomic signature of mESC[56,96]. However, a recent study showed

**Fig. 6 | *Tcf7l1* is required for PE lineage formation in embryos. a** Regulon activity analysis of TCF/LEF factors during embryo preimplantation development[82]. Circle size indicates % of cells in which TCF/LEF regulons are active. **b** Violin plot depicting Tcf7l1 and Tcf7l2 regulon activity at different developmental stages of embryo development[82]. Dots represent individual cells (*n*). E2.5-E2.75: *n* = 114, E3.0-E3.5: *n* = 146, TE: *n* = 115, EPI: *n* = 71, PE: *n* = 81. Horizontal line denotes the median value, box indicates the interquartile range (IQR) and whiskers denote the 1.5 × IQR. **c** *Tcf7l1* and *Tcf7l2* gene expression along pseudo-time of PE specification trajectory during the transition between E3.5 to E4.5 developmental stages, compared to gene expression levels of different lineage markers[50]. **d** Representative IF image of pre-implantation embryos upon iCRT3 treatment from 2 independent experiments. Scale bar = 50 μM. **e** Number of NANOG+ and GATA6+ cells (counts) per embryo. Mean ± SEM; Control *n* = 17; iCRT3 *n* = 19; 2 independent experiments. multiple unpaired *t* tests with Holm-Sidak method. **f** Percentage of NANOG+ and GATA6+ cells normalized on total number of ICM per embryo. Mean ± SEM; Ctrl *n* = 17, iCRT3 *n* = 19; 2 independent experiments. **g** Embryo lumen volume reported in pL. Mean ± SEM; Ctrl *n* = 17, iCRT3 *n* = 19 embryos; 2 independent experiments; two-tailed unpaired *t* test. **h** Representative IF image of WT, negative control CRISPR/Cas9 and *Tcf7l1*−/− CRISPR/Cas9 embryos from 3 independent experiments. Scale bar=50 μM. Dashed lines delimitate lumen cavity. **i**–**k** Analysis of WT, negative control CRISPR/Cas9 and *Tcf7l1*−/− CRISPR/Cas9 embryos. Horizontal line denotes the median value, box refers to the 25th to 75th percentiles and whiskers mark min and max values.; WT *n* = 15, negative control CRISPR/Cas9 *n* = 19, *Tcf7l1*−/− CRISPR/Cas9 *n* = 15. 3 independent experiments. **i** Total number of ICM cells. No statistically significant differences using one-way ANOVA test. **j** Number of NANOG+ and GATA4+ cells (counts) per embryo. One-Way ANOVA test. **k** Embryo lumen volume reported in pL. One-Way ANOVA test. Source data for all experiments are provided as a Source data file.

that complete deletion of *β-catenin* slightly increases the expression of PE markers, while maintaining the self-renewal state[97]. We found that *Tcf7l1* overexpression promotes a stronger phenotype than *β-catenin* deletion as it is sufficient to induce EPI to PE lineage conversion, suggesting that either the transcriptional inhibitory force of TCF7L1 prevails over the transcriptional effect exerted by β-catenin[42,62] or TCF7L1 might regulate PE formation in a β-catenin-independent manner[98]. Intriguingly, it has been previously shown that, unlike inhibition of only canonical Wnt pathway, inhibition of all (canonical and non-canonical) Wnt ligand secretion leads to post-EPI differentiation[31]. Although PE differentiation was not assessed, earlier studies and our results might indicate that the non-canonical Wnt pathway might have a prominent role in post-EPI differentiation.

Although our results demonstrate a direct repression of formative genes by TCF7L1 in pluripotent culture conditions, it has been shown that upon naive medium withdrawal, TCF7L1 facilitates formative transition by restraining naive pluripotency network[8]. However, in agreement with previous studies[8,60,99], our results show that deletion of *Tcf7l1* alone delayed but did not abrogate pluripotency exit, EpiSCs or neuroectodermal differentiation. Strikingly, extraembryonic PE differentiation was severely compromised in *Tcf7l1*−/− cells. Furthermore, only the triple KO of *Tcf7l1* in combination with the transcriptional regulators ETV5 and RBPJ was effective in arresting the cells in the pluripotent state, rendering them refractory to differentiation[8]. Interestingly, it has been shown that TCF7L1 can repress definitive endoderm genes and its deletion facilitates endoderm specification at initial stages, when cells are cultured in chemically defined endodermal differentiation medium[100]. This is not in conflict with our results since TCF7L1, as all TCF factors, elicits diverse transcriptional programs in a context- and cell state- dependent manner[101]. For instance, TCF7L2 has been described as the main driver of metabolic zonation of the liver by activating zonal transcription, in contrast to its reported role as transcriptional repressor in mESCs[102].

Here, we provide further evidence demonstrating that distinct TCF factors may regulate independent cellular functions. We demonstrate that deletion or overexpression of *Tcf7l1* has drastic effects on the capacity of mESCs to undergo PE cell lineage conversion, differently from *Tcf7*, which shows no significant effects. We and others have previously shown that the Wnt transcriptional repressor TCF7L1 and the activator TCF7 display unique DNA binding sites[36] leading to the regulation of distinct gene sets in mESCs[36,103]. Explicitly, TCF7 regulates Wnt-dependent cell cycle events by directly binding to cyclin-dependent kinase inhibitors such as *p16Ink4a* and *p19Arf*[36]. In contrast, we are not able to detect binding of TCF7L1 on cell cycle genes while it is present on genes regulating naive and formative pluripotency.

In conclusion, our study unravels new aspects of the mechanism governing EPI vs. PE binary cell-fate decisions as part of the interconnected cascades and gene regulatory networks. Further understanding of TCF7L1 function at single-cell and chromatin accessibility level, will contribute to elucidating the complex circuit of differentiation decisions in the preimplantation embryo and beyond.

## Methods

### Ethics declaration
All the experiments performed were approved by the Ethics Committee at KU Leuven University under the ethical approval codes P170/2019 and by Animal Ethics Committee of Ghent University Hospital (approval number ECD 18-29). B6D2F1 mice were obtained from Charles River Laboratories, Brussels, Belgium.

### Cell culture
Undifferentiated wild-type (WT) murine ESCs, *Tcf7l1*−/− mESCs obtained from B. Merril[92] and *Tcf7*−/− mESCs previously generated in the laboratory[36] were cultured at 37 °C and 5% $CO_2$ on gelatin-coated plates in DMEM (Gibco), 15% fetal bovine serum, 2mM L-glutamine (Gibco), 1X minimal essential medium non-essential amino acids, 1x sodium pyruvate, 1x penicillin-streptomycin, 100 μM β-Mercaptoethanol and 1000 U/ml recombinant murine LIF (Peprotech).

For the evaluation of naive EPI and PE-like populations in Figs. 1a, 3c, d and Supplementary Fig.3b, cells were maintained for at least 3 passages in knockout DMEM (Gibco), 20% knockout serum replacement (KOSR), 2 mM L-glutamine, 1x minimal essential medium non-essential amino acids (Gibco), 1x penicillin/streptomycin (Gibco), 100 μM β-Mercaptoethanol and 1000 U/mL recombinant murine LIF.

For the pharmacological modulation of Wnt pathway in Figs. 2a and 3a, undifferentiated WT ESCs were seeded at a density of 250,000 cells per well in 6-well plates and treated with 200 ng/mL Wnt antagonist Dickkopf 1 (DKK1) (Peprotech) or 10 μM iCRT3 [inhibitor of β-catenin–responsive transcription] (Sigma) for 3 passages in KOSR/LIF conditions.

### Retinoic acid differentiation
Retinoic acid (RA) differentiation experiments were performed as previously described in Semrau et al., 2017. Explicitly, prior to differentiation cells were grown for at least 2 passages in 2i medium plus LIF (2i/L): DMEM/F12 (Gibco) supplemented with 0.5x N2 supplement, 0.5x B27 supplement, 0.5 mM L-glutamine, 20 μg/ml human insulin (Sigma-Aldrich), 1 × 100 U/ml penicillin/streptomycin, 0.5x MEM Non- Essential Amino Acids, 0.1 mM 2-Mercaptoethanol, 1 μM MEK inhibitor (PD0325901, Sigma-Aldrich), 3 μM GSK3 inhibitor (CHIR99021, Sigma-Aldrich), 1000 U/ml mouse LIF (Peprotech). Cells were seeded at a density of $2.5 × 10^5$ per 10 cm dish and grown over night (12 h). The next day cells were washed twice with PBS and medium was replaced with basal N2B27 medium (2i/L medium without inhibitors, LIF and the additional insulin) supplemented with 0.25 μM all-trans retinoic acid (Sigma-Aldrich). Medium was being refreshed every 48H.

### *Tcf7l1* transgene induction

*Tcf7l1*-inducible ESC line was purchased from the NIA Mouse ES Cell Bank[68]. ES cells were cultured on feeder cells in ESC medium containing 0.2 μg/ml doxycycline (Dox) (Sigma). Before transgene induction cells were cultured on gelatin coated dish in medium containing 0.2 μg/ml Dox for 3 days. One day before transgene induction, $1 \times 10^6$ ES cells were plated onto gelatin coated 100 cm² dishes for 24H. Dox was removed by washing the cells three times with PBS at interval of 3H. In the absence of Dox, the recombinant locus expresses TCF7L1 and Venus YFP protein connected via a synthetic internal ribosomal entry site (IRES). Following the 24H of induction (Day 1), cells were dissociated using 0,5% trypsin-EDTA and resuspended in FACS buffer [PBS supplemented with 5% FBS]. Venus[high] expressing cells were sorted by fluorescence-activated cell sorting (FACS) using a BD FAC-SAria™ III sorter and replated in ESC medium conditions. Cells were harvested after 2, 4 and 8 Days after Dox removal.

### EpiLCs differentiation

WT and *Tcf7l1*$^{-/-}$ mESCs were maintained for at least 5 passages in N2B27 plus 2i/LIF. Cells were plated at a density of $1 \times 10^4/cm^2$ in fibronectin (Millipore) coated (16.7 mg/ml) 6-well plates in N2B27 basal medium supplemented with 20 ng/ml activin A (Peprotech), 12.5 ng/ml Fgf2 (Peprotech) and 2 μM XAV939 (Sigma) for 2 and 3 days. For the TCF7L1 overexpressing cells, transgene induction was performed as explained before and Venus[high] expressing cells were FACS-sorted and replated in the same Epi-inducing medium for 2 and 3 days.

### Mouse embryo recovery, culture and pharmacological modulation of Wnt signaling

Mice were pathogen-free, and bred and maintained in the animal care facility at KU Leuven. Animals had access to food (ssniff® R/M-H) and water (HCl-acidified water (pH 2.5-3) and were group-housed in ventilated cages under controlled temperature (22 ± 2 °C) and humidity (45–70%) with a 14 h light, 10 h dark light cycle. To obtain pre-implantation embryos, CD-1 female mice were superovulated (SO) by intraperitoneal injection of 150 μL pregnant mare's serum (PMS) gonadotropin, followed by injection of 150 μL human chorionic gonadotropin (hCG) 48 h later. SO females were then mated with male mice. For β-CATENIN quantification, E3.5 and E4.5 embryos were flushed from dissected uteri with EmbryoMax® M2 Medium (Sigma). Embryos were washed with PBS and fixed in 4% paraformaldehyde (PFA). For ex vivo embryo culture and treatment, E2.5 embryos were flushed from dissected oviducts using EmbryoMax® M2 Medium. E2.5 embryos were treated with 100 ng/mL DKK1 and 10 μM iCRT3 diluted in EmbryoMax® KSOM Medium and cultured in cell culture dish (Nunc™ Cell-Culture Treated Multidishes, Thermo Fisher Scientific) at 37 °C in incubator supplied with 5% CO₂. Blastocysts (E2.5 + 48H) were stopped and fixed in 4% PFA for further molecular analysis (see Immunofluorescence staining section).

### Antibody staining and flow cytometry

For the evaluation of naive EPI and PE-like populations in Fig. 1a, 3c, d and Supplementary Fig. 3b, cells growing as described above were washed once with PBS and then incubated with Cell dissociation solution (Sigma) for 20 min at 37 °C. Cells were washed twice with PBS and were counted. Conjugated antibodies were added in PBS (0,2 μg for each $1 \times 10^6$ cells/1:100) and samples were incubated for 30 min at 4 °C. We used the following antibodies: PE anti-mouse CD140a (PDGFRA) (Thermo Fisher Scientific, 12-1401-81), APC anti-mouse CD31 (PECAM-1) (Thermo Fisher Scientific, 17-0311-82), PE Rat IgG2α (BD Biosciences, 553930), APC Rat IgG2α (Thermo Fisher Scientific, 553930). The antibodies used in this study can be found in Supplementary Table 1. Following antibody incubation, the cells were washed once in FACS buffer, resuspended in fresh FACS buffer, and passed through filter. For the differentiation experiments shown in Fig. 3e–f

and Supplementary Fig. 3c–d, cells growing as described above were first dissociated with Cell dissociation solution (Sigma) for 20 min at 37 °C. Then, cells were incubated in a volume of 250 μl of basal (N2B27) medium with conjugated antibodies (0,2 μg for each $1 \times 10^6$ cells) for 30 min at 37 °C, in the dark. Afterwards, cells were washed once with PBS, resuspended in basal medium and filtered. Flow cytometry was performed using a BD Canto HTS. Unstained and isotype control samples were used for gating on forward and side scatter. Data analysis was carried out using FlowJo software.

### Gene expression analysis

Total RNA was extracted from cells using GenElute™ Mammalian Total RNA Miniprep Kit. cDNA was reverse transcribed from 500 ng of RNA using the iScript cDNA synthesis kit according to manufacturer's guidelines. Real-time quantitative PCR (RT-qPCR) was performed in three technical replicates per sample using SYBR Green master mix on a ViiA 7 Real-Time PCR system (AB applied biosystems) utilizing specific primers at a concentration of 1 μM. Primer sequences used in this study are specified in Supplementary Data 11. Data analysis was performed with QuantStudio™ Real-Time PCR Software. Ct values detected for each sample were normalized to the housekeeping gene *Gapdh*.

### Western blot analysis

ES cells were washed with PBS, trypsinized, and collected by centrifugation. Whole cell lysates were prepared using RIPA cell lysis buffer (Sigma) containing 1:100 phosphatase inhibitor cocktail 2, phosphatase inhibitor cocktail 3 and protease inhibitor (Sigma). Samples were rotated for 30 min at 4 °C and spun at max speed for 10 min at 4 °C. The supernatant from samples was collected and protein concentration was determined by Bradford assay. Equal amounts of protein per sample were combined with Laemmli buffer, denatured for 5 min at 95 °C and subjected to SDS/PAGE separation, followed by immunoblotting. The following primary antibodies were used: rabbit anti-active β-Catenin (Cell Signaling Technology, 8814), mouse anti-total β-Catenin (BD Biosciences, 610154), rabbit monoclonal anti-Tcf1 (Cell Signaling Technology, 2203), rabbit monoclonal anti-Lef1 (Cell Signaling Technology, 2230). Mouse anti-β-ACTIN (Santa Cruz Biotechnology; sc-47778) was used as a load control. The antibodies used in this study can be found in Supplementary Table 1. Protein quantification was performed with ImageJ software from $n = 3$ independent biological replicates (Fig. 1e). Uncropped and unprocessed scans are provided in the Source data file (for Fig. 1e) or in Supplementary Fig. 8 for blots presented in Supplementary figures.

### Immunofluorescence staining

WT and *Tcf7l1*$^{-/-}$ cells were cultured on gelatin-coated glass coverslips for 4 days as described in the Retinoic acid differentiation section. *Tcf7l1*-OE Day0, *Tcf7l1*-OE Day6, XEN cells and embryos were cultured as described above. Cells and embryos were washed 2× in PBS, fixed in 4% PFA for 20 min at room temperature (RT) and permeabilized with 0,2% Triton X-100 in PBS/donkey serum (DS) 3% for 10 min. Samples were then blocked with PBS/DS 5% + 0,2% Tween20 + 0,2% BSA for 30 min at RT and incubated with primary antibodies overnight (o/n) at 4 °C. Primary antibodies were diluted in PBS/DS 3% + 0,02% Tween20 at the specific working concentration. Rat anti-Nanog (eBioscience™, 14-5761-80) (1:200), goat anti-Gata6 (R&D systems, AF1700) (1:200), mouse anti-Cdx2 (Biogenex, AM392GP) (1:200), goat anti-Nestin (Santa Cruz Biotechnology, sc-21248) (1:200), mouse anti-Gata4 (BD Biosciences, 560327) (1:250), rabbit anti-active β-Catenin (Cell Signaling Technology, 8814) (1:200), mouse anti-total β-Catenin (BD Biosciences, 610154) (1:200), goat anti-Tcf3 (Santa Cruz Biotechnology, sc-8635) (Tcf7l1) (1:200). The antibodies used in this study can be found in Supplementary Table 1. Next, samples were repeatedly washed and incubated with secondary-AlexaFluor antibodies (1:500 in PBS/DS 3% + 0,02% Tween) for 1 h at RT. Secondary

antibodies used in this study: Donkey anti-Goat IgG (H + L), Alexa Fluor 488 (Invitrogen), Donkey anti-Rat IgG (H + L), Alexa Fluor 555 (Abcam), Donkey anti-Rabbit IgG (H + L), Alexa Fluor 647 (Abcam), Donkey anti-Mouse IgG (H + L), Alexa Fluor 647 (Invitrogen), Donkey anti-Goat IgG (H + L), Alexa Fluor 647 (Invitrogen). DAPI was used to stain nuclei. Lastly, stained embryos were mounted in 10 µL PBS/DS 3% drops covered with mineral oil on 35 mm glass-bottomed dishes. Embryos were imaged under a Leica SP8x confocal microscope. WT and *Tcf7l1*[−/−] cells were mounted on microscope slides imaged under a Zeiss AxioImager Z1 Microscope using the AxioVision SE64 software. *Tcf7l1*-OE Day0, *Tcf7l1*-OE Day6, XEN cells were imaged under an Operetta CLS™ high-content analysis system. Image quantification and analysis was performed using ImageJ software and Harmony High-Content Imaging and Analysis software.

## Sample preparation and RNA sequencing
RNA extraction was performed using GenElute™ Mammalian Total RNA Miniprep Kit using On-Column DNase I Digestion Set. RNA was quantified with Nanodrop 1000 spectrophotometer (Thermo Fisher Scientific) and analyzed using a 2100 Bioanalyzer (Agilent Technologies). Three biological replicates were prepared for each sample. Libraries were produced using KAPA Stranded RNA-Seq Library Preparation Kit Illumina® Platforms. Final cDNA libraries were checked for quality using a 2100 Bioanalyzer and quantified using Qubit dsDNA HS Assay Kit. The libraries were loaded in the flow cell at 4 nM concentration and sequenced on the Illumina HiSeq4000 sequencing system producing single-end 50-nucleotide reads at Genomics Core Leuven (Belgium).

## Quality assessment and mapping
Adapters were removed using Cutadapt and processed reads shorter than 20 bp were discarded. Mapping was performed with HISAT2 v2.1.0, against the mouse reference genome (mm10) with default parameters. Alignment scores (percentages of uniquely mapped reads) ranged from 71.2% to 81.5%. A grand total of >340Mi uniquely mapped reads was used for further analyses.

## Gene assignment and differential gene expression analysis
FeatureCounts v2.0.1[104] was used to count mapped reads for genomic features in the mouse transcriptome (Ensembl Release M25, GRCm38.p6). A total of 18584 genes with average count >5 were kept and passed to DESeq2 v1.28.1[105] for the detection of statistically significant differentially expressed genes (Supplementary Data 4; Supplementary Data 5). Differentially expressed genes were called on the basis of DESeq2 thresholds for absolute log2-fold-change (≥2) and a 5% false discovery rate (Fig. 5b–d).

## Functional enrichment analysis
Functional enrichment analysis of gene expression was performed with a combination of the Cytoscape ClueGO plug-in and a custom R function based on the igraph library. Differentially expressed gene lists were analyzed with ClueGO. Community detection was applied on the resulting network using the Louvain algorithm, while also taking into account edge betweenness centrality of the functional terms. Network communities were associated to functional terms on the basis of term name over-representations and color-coded accordingly (Fig. 5a; Supplementary Data 6; Fig. 5e; Supplementary Data 7). Gene enrichments at the level of transcriptional regulation were calculated against the TTRUST database using Metascape[81] (Supplementary Fig. 5a).

## TCF7L1 binding analysis
Information for TCF7L1 binding was obtained from publicly available ChIP-seq data (E-MTAB-4358)[36]. TCF7L1 peaks were called using GEM as described in De Jaime-Soguero et al.[36]. Reads were aligned to the latest mouse genome available (mm10, Genome Reference Consortium GRCm38) and genes with TCF7L1 peaks within 10 kb around the transcription start site (TSS) were considered to be bound by TCF7L1. Functional enrichment on the intersection of TCF7L1-bound and day1 differentially downregulated genes was performed with gProfiler (Fig. 5h)[106]. All analyses of gene expression were performed in R. Visualizations of TCF7L1 binding data were made with the use of the IGV Genome Browser[107].

## Gene regulatory network analysis using SCENIC
Gene regulatory network analyses in Fig. 6a,b were performed using publicly available resource dataset from Posfai et al.[82] (GSE145609). Gene regulatory network inference was performed using pySCENIC v0.9.15[108] and regulon activity in samples from Mohammed et al.[49] dataset (GSE100597) was extracted from the integrated single-cell mouse preimplantation development atlas[82]. Visualization was performed using DotPlot and and VlnPlot functions in Seurat v3.0.0[109].

## Single-cell RNAseq analysis
Single-cell RNA-seq analyses shown in Fig. 6c, Supplementary Figs. 1b and 6a were performed using publicly available dataset and code as previously described in Nowotschin et al.[50] (GSE123046). Briefly, raw read counts from two replicate samples of the mouse embryonic development at E3.5 and E4.5 stages were used as an input for Harmony algorithm v0.1.4[50]. Raw reads were normalized to the library size and log transformed, highly variable genes were used to construct Harmony augmented affinity matrix which was then used to generate a visualization with a force directed layout (Supplementary Figs. 1b and 1c). The differentiation trajectory was inferred using Palantir algorithm v1.0.0[110] and Harmony augmented affinity matrix. Branches were considered PE or EPI if the corresponding branch probability was higher than 0.9 (Supplementary Fig. 1b; Fig. 6c and Supplementary Fig. 6a).

Gene expression analysis in Fig. 1g and Supplementary Fig. 1e was performed using published datasets[49,50]. Normalized read counts were used as input for Seurat DoHeatmap function[109].

## Ovarian stimulation and zygote collection
B6D2F1 hybrid female mice (6–12 weeks old, Charles River Laboratories, Belgium) underwent ovarian stimulation, by intraperitoneal injection of 5IU pregnant mare's serum gonadotropin (PMSG, Folligon®, Intervet, Boxmeer, Netherlands), and 5IU human chorionic gonadotrophin (hCG) 48 h apart, after which natural mating was allowed o/n. Zygotes were removed from the ampulla of the oviduct, followed by 3-min incubation in 200 IU/ml hyaluronidase, and several washing steps in KSOM medium supplemented with 0.4% bovine serum albumin (Millipore).

## CRISPR/Cas9 zygote microinjection and embryo culture
*Tcf7l1*-targeting CRISPR RNA (crRNA) (protospacer sequence 5′-GGTCTGGAATCATCAGGAAG − 3′, custom, Integrated DNA Technologies, Belgium), negative control crRNA (1072544, Integrated DNA Technologies, Belgium), and trans-activating crRNA (tracrRNA) (1072533, Integrated DNA Technologies, Belgium), were dissolved in duplex buffer (1072547, Integrated DNA Technologies, Belgium). Sequence is also provided in Supplementary Data 11. A crRNA:tracrRNA complex (guide RNA, gRNA) was formed by mixing the crRNA and tracrRNA in a 1:1 molar ratio, followed by 5 minutes incubation at 95 °C. Once the mixture cooled down to room temperature, Cas9 protein (1072545, Integrated DNA Technologies, Belgium) was added to the mixture to form a ribonucleoprotein (RNP), after which the RNP was further diluted in Gamete Buffer, to a final concentration of 25 ng/µl gRNA and 50 ng/µl Cas9, aliquoted and stored at −80 °C until further use. Zygotes were piezo-electric microinjected in KSOM medium supplemented with 20% fetal bovine serum (Gibco BRL, Gaithersburg, USA) by aspirating an amount of the RNP complexes, equal

to the zygote diameter. Following injection, mouse embryos were cultured until the 8-cell stage in KSOM medium supplemented with 0.4% bovine serum albumin (Millipore), after which the medium was changed to Cook Blasto ® (Cook Ireland Ltd, Ireland). Incubation took place at 37 °C, 5% $O_2$ and 6% $CO_2$ in all cases.

### DNA extraction and sequence analysis

DNA was extracted from single fixed and immune-stained embryos with Arcturus picopure™ DNA extraction kit in a volume of 10 μl extraction solution, according to the manufacturer's instructions. Extracted DNA was PCR-amplified (F 5′ – GTGCCTTCTCCGTCAGTCTC – 3′, R 5′ – GCAGGCACAAATCCAAGTTT – 3′, and subsequently subjected to next-generation sequencing in an Illumina MiSeq platform[111]. Primer sequences are also provided in Supplementary Data 11. Raw sequencing data were analyzed by the BATCH-GE tool[112].

### Image analysis and quantification

Embryo imaging was performed on Leica SP8x confocal microscopy with precise galvo-Z for quick Z-stack imaging. To quantify total and nuclear active (non-phosphorylated) β-Catenin signals, the cells of NANOG- and GATA6-positive cells were manually selected by using ImageJ software from single 2D confocal images. The total active (membrane + cytoplasmatic+ nuclear) and nuclear active (non-phosphorylated) β-Catenin signal levels were measured by ImageJ software. Total active β-Catenin signal in NANOG$^+$ and GATA6$^+$ cells was calculated selecting both membrane and nuclear signals. Nuclear active β-Catenin signal in NANOG$^+$ and GATA6$^+$ cells was calculated selecting specifically the DAPI signal area. Embryos were imaged using the same imaging parameters (laser power and smart gain). Images were merged and/or quantified using ImageJ software. Embryo lumen volume measurement of embryo treated with DKK1, iCRT3 and $Tcf7l1^{-/-}$ was performed by analyzing blastocyst z-stacks obtained using Leica SP8x confocal microscopy with precise galvo-Z for quick Z-stack imaging. The total length of the z-stack was divided by the total number of frames per embryo. Therefore, a constant number representing the height of the z-stack step size was identified using the Leica Application Suite X (LAS X, v3.7.4) software. The lumen area of each frame was calculated using ImageJ. Afterwards, lumen area was multiplied by the height of the z-stack step size. The summary provides the lumen volume of the whole embryo in mm3. Volume was then converted to pico liter (pL) by dividing the lumen volume in mm3 by 1000. Cell fate specification analysis was performed by analyzing z-stacks from embryos, which were immunostained for cell lineage-specific markers, CDX2 (TE), GATA6/GATA4 (PE) and NANOG (EPI). For Fig. 2c–e and Supplementary Fig. 2b, embryos were analyzed for GATA6 and CDX2 immuno-colocalization and only GATA6$^+$/CDX2$^-$/NANOG$^-$ were classified as PE (indicated as GATA6$^+$ in the figure). NANOG$^+$/GATA6$^-$ cells were classified as EPI (indicated as NANOG$^+$ cells in the figure). CDX2$^+$ were classified as TE. Cell nuclei were identified by DAPI staining.

### Statistical analysis

All statistical analyses in this study were performed using GraphPad Prism 6 software (San Diego, CA, USA) and R Project for Statistical Computing (*R v3.6.1*). Statistical significance was evaluated by unpaired two-tailed Student's *t*-test or Mann–Whitney *U* test where appropriate. Significant differences were determined by one-way and two-way analyses of variance ANOVA for multiple groups or for more than one variance, respectively. Data are presented as mean ± standard error of mean (SEM) or standard deviation (SD), and as Log2 of Fold change (FC). Data presented as violin plots show median and quartiles with individual data points.

### Reporting summary

Further information on research design is available in the Nature Portfolio Reporting Summary linked to this article.

## Data availability

All data generated in this study are provided in the article file, Supplementary Information, Supplementary Data and Source Data files. The RNA-seq data generated in this study has been deposited is the Gene Expression Omnibus (GEO) database under the accession code: GSE171204. The publicly available TCF7L1 ChIP-seq data in mESCs used in this study are available in the ArrayExpress database under the accession code: E-MTAB-4358. The publicly available scRNA-seq data used for gene expression and SCENIC analysis are available in the GEO database under the accession codes: GSE145609, GSE100597 and GSE123046. Source data are provided with this paper.

## Materials availability

Requests for materials should be addressed to Frederic Lluis (Frederic.lluisvinas@kuleuven.be) and Paraskevi Athanasouli (paraskevi.athanasouli@kuleuven.be).

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

## Acknowledgements

We would like to thank Dr. Brad Merrill for providing the WT and *Tcf7l1⁻/⁻* mESCs cells and Samantha Zaunz for helping on FACS experiments. We are grateful to Susan Schlenner and the KU Leuven FACS core team for providing the facility and especially Reena Chinaraj who helped us during flow cytometry experiments. We also thank Lotte Schoeters and Alvaro Cortes Calabuig from the KULeuven Genomics Core (http://genomicscore.be) for RNA sequencing, data processing and analysis. The authors would like to extend their gratitude to the FWO Research Foundation – Flanders for the Ph.D. fellowships awarded to P.A. (11M7822N), B.V. (11E7920N), A.J. (1158318 N), postdoctoral funding to A.B. (1298722 N), KU Postdoctoral Mandate awarded to ADJS (PDM/18/212) and FWO-Vlaanderen Research Project Grants G097618N, G091521N (F.L.L.), G073622N (F.L.L., B.H.), G092518N, G0C6820N (K.P.K.), G0C9320N, G0B4420N (V.P.), EOS grant G0I7822N (V.P.), Ministerio de Ciencia e Innovación 008506-PID2020-114080GB-I00 (M.P.C.) and AGAUR grant 006712 2017-SGR 689 (M.P.C.) and C1 KU Leuven internal grants C14/21/115 (F.L.L.), C14/21/119 (V.P.).

## Author contributions

P.A. conducted the experimental set up, work and analysis of the following, the iCRT3 treatment, the RA differentiation, the Tcf7l1 and Tcf7 overexpression, the EpiLCs differentiation, performed the qRT-qPCRs, the FACS plot analysis, the western blots, immunofluorescence analysis, the Tcf7l1 OE sorting experiments and obtained the samples used for RNA-seq analysis, prepared RNA-seq libraries, performed the experiments during revision process including DKK1 treatment on mouse embryos (developmental analysis, lineage evaluation) and β-catenin levels evaluation in fresh E4.5 isolated embryos, reanalysis of previously obtained data, prepared final figures and the manuscript related files, wrote the manuscript. M.B. performed bioinformatic analysis on published data, prepared RNA-seq libraries, performed immuno-fluorescence analysis on Tcf7l1-OE-mESCs and for β-catenin levels at E3.5 and E4.5, performed mouse embryo experiments and analysis for lineage evaluation, lumen volume upon DKK1 and iCRT3 treatment, and of Tcf7l1–/– embryos, prepared figures and participated in the initial writing of the manuscript. A.d.J.S. participated in the conception of the project, performed DKK1 treatment experiment on mESCs and EPI/PE populations sortings, contributed to characterize the EPI/PE sub-populations in WT, *Tcf7l1 ⁻/⁻* and *Tcf7 ⁻/⁻* cells (Figs. 1a, 3c–d, Suppl. Fig. 3b) under basal conditions, participated in the initial experimental set up and performed Tcf7l1-OE experiments for sorting, performed qRT-qPCR (Fig. 2b, Suppl. Fig. 4c), FACS analysis (Fig. 2a), western blot analysis (Suppl. Fig. 1a), provided advice and comments on the manuscript. A.B. performed the CRISPR components design, zygote injection and analysis of MiSeq results. S.P. analyzed the RNA-seq data generated in this study and the published binding data and produced related figures. B.K.V.D.V. helped with mouse embryos experiments. A.J. analyzed the published sc-RNAseq datasets from mouse embryos and produced the related figures. T.V. performed the analysis of mouse embryos for lineage evaluation upon DKK1 treatment during revision. A.F. helped with mouse embryos experiments. Y.E.L. helped with sorting experiments. A.L.N., F.A., K.P.K., V.P., M.P.C., C.V., A.Z., and B.H., provided advice and comments on the manuscript. C.N. performed analysis on the RNA-seq data generated in this study and the published binding data and produced figures, helped with statistical analysis, provided advice and comments on the manuscript. F.L. conceived the project, guided the research, interpreted data and wrote the manuscript. All authors edited the manuscript and agreed on its final version.

## Competing interests

The authors declare no competing interests.

## Additional information

[1]KU Leuven, Department of Development and Regeneration, Stem Cell Institute, B-3000 Leuven, Belgium. [2]Ghent-Fertility And Stem cell Team (G-FaST), Department for Reproductive Medicine, Department for Human Structure and Repair, Ghent University Hospital, 9000 Ghent, Belgium. [3]Department of Rheumatology, Clinical Immunology, Medical School, University of Crete, 70013 Heraklion, Greece. [4]Computational Genomics Group, Institute of Bioinnovation, Biomedical Sciences Research Center "Alexander Fleming", 16672 Athens, Greece. [5]Department of Cardiovascular Sciences, KU Leuven, 3000 Leuven, Belgium. [6]Centre for Genomic Regulation (CRG), Dr Aiguader 88, 08003 Barcelona, Spain. [7]KU Leuven Institute for Single-Cell Omics (LISCO), 3000 Leuven, Belgium. [8]ICREA, Pg. Lluis Companys 23, Barcelona 08010, Spain. [9]Universitat Pompeu Fabra (UPF), Barcelona, Spain. [10]These authors contributed equally: Paraskevi Athanasouli, Martina Balli. ✉e-mail: anchel.dejaime@cos.uni-heidelberg.de; Frederic.lluisvinas@kuleuven.be

