## [Peer Review File · Nature Communications]

The Wnt/TCF7L1 transcriptional repressor axis drives primitive endoderm formation by antagonizing naive and formative pluripotencyEditorial Note: This manuscript has been previously reviewed at another journal that is not operating a transparent peer review scheme. This document only contains reviewer comments and rebuttal letters for versions considered at *Nature Communications* .

REVIEWER COMMENTS

Reviewer #1 (Remarks to the Author):

In the submitted paper, the authors investigate the mechanisms that allow mouse embryonic stem cells (mESCs) to differentiate into epiblast (EPI) or primitive endoderm (PE) cells. By using loss and gain of function studies, performed in vivo as well as in ex vivo manipulated mouse embryos, the authors identify a previously neglected role of the WNT transcriptional repressor TCF7L1 in promoting the PE fate.

The findings are new and of great importance for our understanding of how mESCs make their choice of maintaining pluripotency or committing into their first differentiation event. The article is well written, and the results clearly presented and integrated within the existing knowledge.

This study bears the additional merit of balancing in vitro experiments with valuable in vivo counterpart setups, persuading that the findings are of physiological relevance.

Finally, having benefited of an already existing revision, I wish to point out that several of the points raised by the two Referees – together with the responses provided by the authors – abrogated most of my own concerns. Also, I commend the work that the authors performed to improve their study.

Below I present a few comments that, in my opinion, could improve the clarity of the study if addressed. All of them should be considered of minor nature.

1) In the abstract, the authors write that “TCF7L1 promotes pluripotency exit and suppresses epiblast lineage formation, thereby driving cells into PE specification”. I find this statement lackluster and not representative of the evidence, as it sounds as if PE fate happens as a “passive” consequence of inhibiting pluripotency and the default EPI fate. However, several pieces of evidence presented indicate something stronger, namely that TCF7L1 actively promotes PE lineage commitment. In particular, that Tcf7l1 KO causes a dramatic loss (rather than a quantitative decrease) of PE-like cells upon RA treatment (Fig 3) seems to indicate that TCF7L1 is required to make this lineage commitment.

2) Line 56: “activation of Wnt canonical pathway” should be substituted with “GSK3 inhibition”, perhaps specifying that among the effects of CHIR relevant is β -catenin stabilisation. I would be cautious, at least in the introductory paragraph, as growing evidence indicate that CHIR is required even when β -catenin is genetically deleted, suggesting that pluripotency per se can be maintained in the absence of an active canonical Wnt/ β -catenin signaling (PMID: 34525377; PMID: 32822589).

3) In a few instances the work of Berge et al., 2011 is referenced when commenting the requirement of Wnt to prevent EPI differentiation. In this light, the observation that TCF7L1 favours the PE state strikes me as surprising, especially if the results are interpreted based on current models where TCF7L1 represses the Wnt/ β -catenin canonical pathway. In this case, shouldn't TCF7L1 mostly drive cells to the EPI state? This means that one of the following three possibilities must be true: i) TCF7L1 has a function that goes beyond its role as inhibitor of canonical Wnt/ β -catenin signalling (and this would be consistent with other studies, e.g., PMID: 34525377); ii) it is non canonical Wnts that, when blocked, lead to EPI differentiation (PMID: 21841791); iii) Berge et al. have not addressed PE differentiation (they would have found it if they did). I personally favour option one (i), as TCF7L1 OE not only pushes cells toward PE (instead of EPI, unlike inhibition of all Wnts secretion), but also reduces the expression of pluripotency genes likely in a manner that it is independent of upstream activation of the Wnt pathway. Moreover, iCRT3 addition induced an increase of GATA6 positive cells:

one could speculate that this inhibitor, which should also interfere with the interaction between β -catenin and TCF7L1, yet it does not impair PE cell fate, provides evidence that while PE choice over EPI must require Wnt/ β -catenin inhibition, PE differentiation is ultimately driven by TCF7L1 in a β -catenin independent fashion.

I find the distinction of these possibilities worth mentioning, as it might underlie a novel mechanism of regulation by TCF7L1 that does not necessarily imply canonical Wnt signalling.

4) In Fig 1f-g, the authors comment that EPI cells display high Wnt, as they express abundantly Axin2 and Myc, among others, whilst PE cells have lower active signalling, as they express Gsk3 and Dkk1, which are negative regulators of the pathway. While the evidence presented has convinced me that EPI have higher Wnt than PE (in particular the level of nuclear, activated β -catenin), I suggest the authors to revisit the statement in lines 124-125: in fact, both Axin2 and Dkk1 are Wnt target genes (https://web.stanford.edu/group/nusselab/cgi-bin/wnt/target_genes), and they both are negative regulators of the pathway! This is a double standard of logic, and I suggest being more careful in the description of this phenomenon.

5) Line 145: I do not see the need of specifying "paracrine". By DKK1 administration, also autocrine Wnt should be blocked. Could the authors perhaps specify more clearly what their point is?

6) TCF7L1 OE cells contribute in vivo to PE/VE components. However, the co-expression of dsRed and GATA6 seen at 4 days after Dox-removal and injected into E3.5 blastocysts is not surprising. The authors had already shown that TCF7L1 OE cells start expressing GATA6 at 48h after induction. While it really seems that these cells contribute to the blastocysts development, and their anatomical position is consistent with that of PE, I wonder: do the authors looked at later stages to determine if dsRed positive cells contribute in forming a functional PE (and VE in particular), by expressing broadly PE markers at a later stage, or leading to a live and well embryo? I do not think that this experiment, if not performed or if of difficult execution should delay the publication of this article, but I consider this an important point to raise.

7) Line 308: I believe that TCF7L1 should not be defined as a barrier to formative and EPI differentiation, as TCF7L1 OE still allows upregulation of formative gene, even if to a lower extent (Fig 5i). The conclusion that I would draw, in line with my comment 1, is that TCF7L1 induces/pushes towards a PE fate.

Reviewer #2 (Remarks to the Author):

Athanasouli et al. analyze the effects of Wnt signaling through Tcf7l1 during epiblast and primitive endoderm segregation in the pre-implantation embryo and during mESC differentiation. In contrast to a number of previous publications, they find that Wnt signaling does play a role in EPI/PrE segregation: active Wnt is characteristic of the EPI lineage and Tcf7l1 (a repressor, inactivated by Wnt) is needed in the PrE lineage.

In addition, using mESCs as an in vitro model they show that Tcf7l1 acts as an inhibitor of the pluripotency network and thereby promotes PrE fate.

Overall, there seems to be a minor change in Epi and PrE cell fates in the preimplantation embryo due to perturbations to Wnt/Tcf7l1 (Dkk1 treatment, iCRT3 treatment or Tcf7l1 knock-out). This may be in line with the subtle differences in beta-catenin intensities observed between NANOG and GATA6 positive cells in the embryo.

Main points:

-Trophectoderm cells are not classified by staining for a specific marker. Lingering GATA6 expression

in the TE could therefore falsely be interpreted as an increase in PrE cells (a good example of this is the Dkk1-treated embryo in Figure 2c).

-The chimera experiment is not convincing at all. The signal looks very much like background, and there are no positive or negative controls. Using a cell line with a nuclear-localized fluorescent marker would help distinguish background vs real cell integration.

-The role of Fgf4/Erk (which is well known for being involved in establishing EPI and PrE cell fates, and which when manipulated results in a complete change in cell fate in all ICM cells) is largely ignored in this paper. How would Wnt/Tcf7l1 interact with the Fgf4/Erk pathway?
For example, if MAPK signaling is inhibited, could Wnt inhibition still drive PrE fate? Or conversely, if exogenous Fgf4 is added, could Tcf7l1 deletion still result in EPI cells? Would a Tcf7l1 pulse occur when Fgf4/Erk signaling is inhibited?

Based on the sentence below, do the authors propose that Wnt inhibition is responsible for the initiation of GATA6 expression in the early embryo?

"However, Fgf4^{-/-} embryos express Gata6, which cannot be maintained after E3.25. This suggests that the onset of the PE program is FGF4-independent." If so, this should be demonstrated by performing stainings for GATA6 in E3.25 embryos where Tcf7l1 is knocked out.

In summary, no clear model is proposed, nor tested for how the Wnt and MAPK pathways would interact during EPI/PrE fate specification.

-The change in lumen volume due to Wnt perturbations is interesting, however there is no evidence this has anything to do with the ICM. In fact, it is much more likely due to changes in the TE lineage. Here again, the manuscript would benefit from analysis of TE cell numbers and TE lineage specific marker expression.

The reference used for the sentence below shows that lumen formation facilitates the EPI/PrE fate decision, but does not state anything about cell fate having an effect on lumen formation. "A direct correlation between PE specification and blastocyst lumen expansion has been recently shown."

-The sentence below has two inaccuracies. 1) Fgf4/Erk signaling is not regarded as a maintenance pathway for EPI and PrE cell fates. It is needed for the emergence of the salt-and-pepper NANOG and GATA6 expression pattern in the ICM from an initially co-expressing state. 2) Moreover, it is now well accepted that Fgfr2 is not the main receptor in the early embryo, but rather Fgfr1 is, which is expressed in all ICM cells.

"In detail, FGF/MAPK is required for the maintenance of PE cells in the ICM as a result of the inverse expression of Fgf4/Fgfr2 in EPI-fated and PE-fated cells respectively."

-The discussion on the origin of Wnt ligands is unclear to me. To explain why the Porcn^{-/-} mutants do not have a pre-implantation phenotype, do the authors suggest that exogenous Wnt ligands, supplied by the maternal environment, are responsible for differentially activating Wnt in EPI and PrE cells? In this case, how would the authors explain that embryos cultured in minimal media (without any Wnt ligands) can specify the correct number of EPI and PrE cells?

Minor:

-Specification and commitment are used interchangeably in the manuscript.

REVIEWER

COMMENTS

Reviewer #1 (Remarks to the Author):

In the submitted paper, the authors investigate the mechanisms that allow mouse embryonic stem cells (mESCs) to differentiate into epiblast (EPI) or primitive endoderm (PE) cells. By using loss and gain of function studies, performed in vivo as well as in ex vivo manipulated mouse embryos, the authors identify a previously neglected role of the WNT transcriptional repressor TCF7L1 in promoting the PE fate.

The findings are new and of great importance for our understanding of how mESCs make their choice of maintaining pluripotency or committing into their first differentiation event. The article is well written, and the results clearly presented and integrated within the existing knowledge.

This study bears the additional merit of balancing in vitro experiments with valuable in vivo counterpart setups, persuading that the findings are of physiological relevance.

Finally, having benefited of an already existing revision, I wish to point out that several of the points raised by the two Referees – together with the responses provided by the authors – abrogated most of my own concerns. Also, I commend the work that the authors performed to improve their study.

Below I present a few comments that, in my opinion, could improve the clarity of the study if addressed. All of them should be considered of minor nature.

We would like to thank the reviewer for the constructive criticisms and suggestions that helped clarify and improve the article's conclusions.

1) In the abstract, the authors write that “TCF7L1 promotes pluripotency exit and suppresses epiblast lineage formation, thereby driving cells into PE specification”. I find this statement lackluster and not representative of the evidence, as it sounds as if PE fate happens as a “passive” consequence of inhibiting pluripotency and the default EPI fate. However, several pieces of evidence presented indicate something stronger, namely that TCF7L1 actively promotes PE lineage commitment. In particular, that *Tcf7l1* KO causes a dramatic loss (rather than a quantitative decrease) of PE-like cells upon RA treatment (Fig 3) seems to indicate that TCF7L1 is required to make this lineage commitment.

Our Chromatin-immunoprecipitation (ChIP-seq) analysis integrated with RNA-seq analysis of downregulated DEG upon TCF7L1 OE detected a significant binding of TCF7L1 and consequent transcriptional repression of epiblast (pluripotent) genes. Therefore our results indicate that TCF7L1 actively promotes epiblast transcriptional repression, which consequently induces PE phenotype. The possibility that extraembryonic endoderm (ExEn) fate differentiation is determined by repression of the pluripotent/Epiblast (EPI) transcriptional program has been previously hypothesized, as disruption or elimination of key EPI regulators' expression, *Nanog* or *Prdm14*, induces ExEn differentiation (Ma et al, 2010, Mitsui et. al, 2003). Our results are more in line with this hypothesis.

However, we agree with the reviewer that TCF7L1 expression is necessary for PE specification. Following the reviewer's suggestion, we have included this sentence in the abstract:

Line 31:

Conversely, TCF7L1 is required for PE specification as deletion of *Tcf7l1* abrogates PE differentiation without restraining epiblast priming.

2) Line 56: “activation of Wnt canonical pathway” should be substituted with “GSK3 inhibition”, perhaps specifying that among the effects of CHIR relevant is β -catenin stabilisation. I would be cautious, at least in the introductory paragraph, as growing evidence indicate that CHIR is required even when β -catenin is genetically deleted, suggesting that pluripotency per se can be maintained in the absence of an active canonical Wnt/ β -catenin signaling (PMID: 34525377; PMID: 32822589).

The reviewer is completely right. We substituted the sentence as suggested by the reviewer.

Line 56:

Selective inhibition of the FGF/ERK pathway together with GSK3 inhibition in mESC culture, using so-called 2i medium, promotes ground-state pluripotency^{20,21}, in which the PE-like population is absent^{13,20,22}.

3) In a few instances the work of Berge et al., 2011 is referenced when commenting the requirement of Wnt to prevent EPI differentiation. In this light, the observation that TCF7L1 favours the PE state strikes me as surprising, especially if the results are interpreted based on current models where TCF7L1 represses the Wnt/ β -catenin canonical pathway. In this case, shouldn't TCF7L1 mostly drive cells to the EPI state? This means that one of the following three possibilities must be true: i) TCF7L1 has a function that goes beyond its role as inhibitor of canonical Wnt/ β -catenin signalling (and this would be consistent with other studies, e.g., PMID: 34525377); ii) it is non canonical Wnts that, when blocked, lead to EPI differentiation (PMID: 21841791); iii) Berge et al. have not addressed PE differentiation (they would have found it if they did). I personally favour option one (i), as TCF7L1 OE not only pushes cells toward PE (instead of EPI, unlike inhibition of all Wnts secretion), but also reduces the expression of pluripotency genes likely in a manner that it is independent of upstream activation of the Wnt pathway. Moreover, iCRT3 addition induced an increase of GATA6 positive cells: one could speculate that this inhibitor, which should also interfere with the interaction between β -catenin and TCF7L1, yet it does not impair PE cell fate, provides evidence that while PE choice over EPI must require Wnt/ β -catenin inhibition, PE differentiation is ultimately driven by TCF7L1 in a β -catenin independent fashion.

I find the distinction of these possibilities worth mentioning, as it might underlie a novel mechanism of regulation by TCF7L1 that does not necessarily imply canonical Wnt signalling.

The reviewer has raised an excellent point. Actually, the dependence or independence of TCF7L1 on the Wnt/ β -catenin pathway has been a matter of discussion in the lab.

Given that treatment of mESCs or embryos with DKK1 (a physiological Wnt canonical inhibitor) increased PE markers expression and the number of PE cells, respectively (Figure 2), we do not feel that we can strongly hypothesize that TCF7L1 effects are completely independent of canonical Wnt pathway. However, the reviewer is completely right in pointing out that the effects of TCF7L1 overexpression are much stronger than DKK1 treatment suggesting that TCF7L1 might have dependent but also independent Wnt functions.

We agree with the reviewer that this hypothesis (as well as the fact that PE formation was not addressed in Ten Berge article) should be mentioned in the discussion. We have introduced them in **line 423-430**.

Line 423-430:

We found that *Tcf7l1* overexpression promotes a stronger phenotype than β -catenin deletion as it is sufficient to induce EPI to PE lineage conversion, suggesting that either the transcriptional inhibitory force of TCF7L1 prevails over the transcriptional effect exerted by β -catenin^{43,63} or TCF7L1 might regulate PE formation in a β -catenin-independent manner¹⁰⁴. Intriguingly, it has been previously shown that, unlike inhibition of only canonical Wnt pathway, inhibition of all (canonical and non-canonical) Wnt ligand secretion leads to post-EPI differentiation³². Although PE differentiation was not assessed, earlier studies and our results might indicate that the non-canonical Wnt pathway might have a prominent role on post-EPI differentiation.

4) In Fig 1f-g, the authors comment that EPI cells display high Wnt, as they express abundantly Axin2 and Myc, among others, whilst PE cells have lower active signalling, as they express Gsk3 and Dkk1, which are negative regulators of the pathway. While the evidence presented has convinced me that EPI have higher Wnt than PE (in particular the level of nuclear, activated β -catenin), I suggest the authors to revisit the statement in lines 124-125: in fact, both Axin2 and Dkk1 are Wnt target genes (https://web.stanford.edu/group/nusselab/cgi-bin/wnt/target_genes), and they both are negative regulators of the pathway! This is a double standard of logic, and I suggest being more careful in the description of this phenomenon.

The reviewer is right. Our main evidence that EPI and PE cells (in the heterogenous mESC population or in blastocysts) show differential Wnt activity levels is evidenced by the significant differential levels of β -catenin. Several transcriptional Wnt targets (such as Axin2 and Dkk1) are both target genes and members of the negative feedback loop. Therefore, classifying them in Wnt target genes or Wnt inhibitors is challenging. We agree with the reviewer and to avoid any misunderstanding we have simplified the categorization into only Wnt positive and negative regulators. To avoid any confusion, we have deleted the Wnt targets category from the text and the figures (see new Fig. 1g).

Line 123:

EPI cells expressed higher levels of canonical Wnt positive regulators (*Fn1*, *Lef1* and *Lgr4*) compared to PE cells. By contrast, PE cells expressed higher levels of genes involved in Wnt pathway repression (*Gsk3b*, *Znfr3* and *Dkk1*) (Fig. 1g and Supplementary Fig. 1e).

5) Line 145: I do not see the need of specifying “paracrine”. By DKK1 administration, also autocrine Wnt should be blocked. Could the authors perhaps specify more clearly what their point is?

In that point, we mean the role of “external administration of DKK1”. In other words, we are not investigating the role of cell- or embryo-produced DKK1. However, we agree with the reviewer that the use of paracrine in this sentence is confusing and we have deleted it.

Line 145:

We therefore assessed if modulation of the canonical Wnt signaling pathway would alter the proportion of naive EPI and PE-like cells *in vitro* as well as within the ICM of mouse embryos.

6) TCF7L1 OE cells contribute *in vivo* to PE/VE components. However, the co-expression of dsRed and GATA6 seen at 4 days after Dox-removal and injected into E3.5 blastocysts is not surprising. The authors had already shown that TCF7L1 OE cells start expressing GATA6 at 48h after induction. While it really seems that these cells contribute to the blastocysts development, and their anatomical position is consistent with that of PE, I wonder: do the authors looked at later stages to determine if dsRed positive cells contribute in forming a functional PE (and VE in particular), by expressing broadly PE markers at a later stage, or leading to a live and well embryo? I do not think that this experiment, if not performed or if of difficult execution should delay the publication of this article, but I consider this an important point to raise.

We agree with the reviewer that studying the developmental potential of TCF7L1 OE cells would be an excellent experiment. We have been trying to optimize this experiment for several months (prior blastocyst injection TCF7L1 OE cells need to be sorted, and this considerably reduced the efficiency of the chimera formation). Moreover, in the last months, the technical personnel of the in-house facility for embryo injection has been replaced. This has reduced even further the efficiency of cell injection and chimera embryo formation. We have been discussing this problem with the editor and decided not to include this experiment in the current article, as continuing to try this experiment would considerably delay the publication of the article.

7) Line 308: I believe that TCF7L1 should not be defined as a barrier to formative and EPI differentiation, as TCF7L1 OE still allows upregulation of formative gene, even if to a lower extent (Fig 5i). The conclusion that I would draw, in line with my comment 1, is that TCF7L1 induces/pushes towards a PE fate.

We agree with the reviewer that TCF7L1 pushes towards a PE fate. Our results indicate that this is happening through the repression of the EPI transcriptional program.

We have demonstrated that OE of TCF7L1 restricts the expression of EPI markers in mESCs cultured in a EpiSCs differentiation medium. We agree with the reviewer that “barrier” might be a strong word, and therefore we have replaced it with “restrains”.

Line 307:

These results confirm that TCF7L1 restrains formative and subsequent primed transition, enforcing cells to PE specification.

Reviewer #2 (Remarks to the Author):

Athanasouli et al. analyze the effects of Wnt signaling through Tcf7l1 during epiblast and primitive endoderm segregation in the pre-implantation embryo and during mESC differentiation. In contrast to a number of previous publications, they find that Wnt signaling does play a role in EPI/PrE segregation: active Wnt is characteristic of the EPI lineage and Tcf7l1 (a repressor, inactivated by Wnt) is needed in the PrE lineage. In addition, using mESCs as an *in vitro* model they show that Tcf7l1 acts as an inhibitor of the pluripotency network and thereby promotes PrE fate.

Overall, there seems to be a minor change in Epi and PrE cell fates in the preimplantation embryo due to perturbations to Wnt/Tcf7l1 (Dkk1 treatment, iCRT3 treatment or Tcf7l1 knock-out). This may be in line with the subtle differences in beta-catenin intensities observed between NANOG and GATA6 positive cells in the embryo.

Main points:

-Trophectoderm cells are not classified by staining for a specific marker. Lingering GATA6 expression in the

TE could therefore falsely be interpreted as an increase in PrE cells (a good example of this is the Dkk1-treated embryo in Figure 2c).

We agree with the reviewer and thank him/her for asking about this point.

We have stained 29 embryos (control) and 30 embryos (DKK1) from 3 independent experiments. Embryos have been treated with the WNT inhibitor DKK1 for 48h in culture. At E2.5+48h, embryos were immunostained for cell lineage-specific markers, CDX2 (TE), GATA6 (PE) and NANOG (EPI). Embryos were analyzed for GATA6 and CDX2 immunocolocalization and only GATA6⁺/CDX2⁻/NANOG⁻ were classified as PE (indicated as GATA6⁺ in the figure). NANOG⁺/GATA6⁻ cells were classified as EPI (indicated as NANOG⁺ cells in the figure). CDX2⁺ were classified as TE. In accordance with our previous results, the new experiments and quantifications demonstrate that embryos treated with DKK1 show a significantly higher number of GATA6⁺ (PE) cells with a reduction of NANOG⁺ cell number indicating a lineage switch in the ICM. Interestingly, the number of CDX2⁺ cells did not change compared to the control (see new Fig. 2c-e).

Our results indicate that Wnt inhibition has the main effect on promoting PE formation. We have changed previous Fig. 2c-e, for new figures that include the CDX2 staining (see new figure below). The text has been changed accordingly.

Line 152:

Lineages assessment showed that DKK1-treated embryos contained a significantly increased number of PE (GATA6⁺) cells compared to control embryos. In addition, DKK1-treated embryos showed a strong but statistically non-significant tendency ($p=0.055$) to have fewer EPI cells (NANOG⁺) than untreated embryos. Interestingly, we did not observe a statistically significant difference in trophectoderm (TE) cell number between control and treated embryos (Fig. 2c, d). Although the total number of ICM cells (Supplementary Fig. 2b) remained unchanged, DKK1-treated embryos exhibited an increased GATA6⁺ PE proportion and a decreased NANOG⁺ EPI proportion (Fig. 2e), demonstrating preferential PE specification at the expense of the EPI upon Wnt inhibition.

Fig. 2 Wnt signaling inhibition promotes formation of the PE lineage. a. Representative IF image of NANOG, GATA6 and CDX2 protein signals in E2.5+ 48h control and DKK1 treated embryo. Nuclei were counterstained with DAPI. Scale bar = 50µM. b. Number of NANOG⁺, GATA6⁺ and CDX2⁺ cells (counts) per embryo. Mean ± SEM; Control: n=29; DKK1 n=30; 3 independent experiments. *t* test ** $p<0.01$; ns= non significant. c. Percentage of NANOG⁺ and GATA6⁺ cells normalized on total number of ICM per embryo. Mean ± SEM; Ctrl n=29, DKK1 n=30; 3 independent experiments.

Supplementary Fig. 2b: Total number of ICM cells per embryo. Mean ± SEM; Control: n=29, DKK1: n=30; 3 independent experiments; unpaired *t* test; ns= non significant.

-The chimera experiment is not convincing at all. The signal looks very much like background, and there are no positive or negative controls. Using a cell line with a nuclear-localized fluorescent marker would help distinguish background vs real cell integration.

We agree with the reviewer that the chimera experiments need improvement. However, as answered to reviewer one, the facility of embryo injection has been experiencing several difficulties in obtaining chimera embryos (from our group and also other groups). Prior to blastocyst injection, the TCF7L1 OE cells need sorting and this creates an extra challenge to the experiment.

In accordance with the editor, as we can not provide new experiments in a short time, we have replaced the current images with the images that we had in the previous version of the article (Figure 4h,i). We have also introduced negative controls (**Supplementary Fig. 4i**).

Supplementary Fig. 4i: Representative BF and fluorescence image of E6.5 non-chimeric embryos injected with *Tcf7l1* induced cells (Venus^{high}/dsRed⁺) (upper panel) and E6.5 non-injected embryos (lower panel).

-The role of Fgf4/Erk (which is well known for being involved in establishing EPI and PrE cell fates, and which when manipulated results in a complete change in cell fate in all ICM cells) is largely ignored in this paper. How would Wnt/Tcf711 interact with the Fgf4/Erk pathway?

For example, if MAPK signaling is inhibited, could Wnt inhibition still drive PrE fate? Or conversely, if exogenous Fgf4 is added, could Tcf7l1 deletion still result in EPI cells? Would a Tcf7l1 pulse occur when Fgf4/Erk signaling is inhibited?

Based on the sentence below, do the authors propose that Wnt inhibition is responsible for the initiation of GATA6 expression in the early embryo?

“However, Fgf4^{-/-} embryos express Gata6, which cannot be maintained after E3.25. This suggests that the onset of the PE program is FGF4-independent.” If so, this should be demonstrated by performing stainings for GATA6 in E3.25 embryos where Tcf7l1 is knocked out.

In summary, no clear model is proposed, nor tested for how the Wnt and MAPK pathways would interact during EPI/PrE fate specification.

FGF and Wnt signaling cooperate in several biological processes. As stated by the reviewer, FGF is the main signaling involved in establishing EPI and PE fates. It would be extremely interesting to investigate the relation/interaction between FGF and Wnt (TCF7L1) pathways. However, this would need several new experiments with a considerable delay in the article's publication. In accordance with the Journal, we have decided not to investigate this point in this article revision.

-The change in lumen volume due to Wnt perturbations is interesting, however there is no evidence this has anything to do with the ICM. In fact, it is much more likely due to changes in the TE lineage. Here again, the manuscript would benefit from analysis of TE cell numbers and TE lineage specific marker expression. The reference used for the sentence below shows that lumen formation facilitates the EPI/PrE fate decision, but does not state anything about cell fate having an effect on lumen formation. “A direct correlation between PE specification and blastocyst lumen expansion has been recently shown.”

We thank the reviewer to point out this interesting suggestion. We have performed TE staining and quantification upon DKK1 treatment. Please, see answer to reviewer's first point.

See new Figure 2c-e.

The reviewer is right. Our sentence “A direct correlation between PE specification and blastocyst lumen expansion has been recently shown.” might induce some misunderstanding. We have changed the order of the sentence in the new version of the article.

Line 162:

A direct correlation between blastocyst lumen expansion and PE specification has been recently shown⁵⁹. Our results show that the blastocyst lumen volume of DKK1-treated embryos was 72% larger (Fig. 2f, g), which was associated with a significant increase of 80% in the embryo size area compared to control embryos (Fig. 2h, i and Supplementary Fig. 2d).

-The sentence below has two inaccuracies. 1) Fgf4/Erk signaling is not regarded as a maintenance pathway for EPI and PrE cell fates. It is needed for the emergence of the salt-and-pepper NANOG and GATA6 expression pattern in the ICM from an initially co-expressing state. 2) Moreover, it is now well accepted that Fgfr2 is not the main receptor in the early embryo, but rather Fgfr1 is, which is expressed in all ICM cells. “In detail, FGF/MAPK is required for the maintenance of PE cells in the ICM as a result of the inverse expression of Fgf4/Fgfr2 in EPI-fated and PE-fated cells respectively.”

- 1) We agree with the reviewer that FGF4/ERK signaling is not considered a maintenance pathway for **both** EPI and PE fates. As a matter of fact our initial sentence stated that “FGF/MAPK is required for the maintenance of PE cells in the ICM”, which has been demonstrated by various articles. It has been shown that even though FGF4/ERK is not required to induce *Gata6* expression, it is necessary for its maintenance in the PE-fated cells after the 32-cell stage (Kang et al., 2013; Krawchuk et al., 2013; Ohnishi et al., 2014). Therefore, we can't assess our statement as inaccurate. However, we appreciate that we neglected to mention the most recognized role of FGF4/ERK in the salt-and-pepper distribution of NANOG and GATA6 in the ICM. We have changed this part of the discussion in the new version. **Line 383.**
- 2) The reviewer is right. It was previously considered that EPI and PE fates within the ICM are established due to the inverse expression of FGF4/FGFR2. EPI-biased cells secrete FGF4, which is received by the PE-biased cells and transduced via the FGFR2 receptor, promoting paracrine induction of PE fate (Guo et al. 2010, Kang et al. 2013). While this is not inaccurate, it is partially incomplete since an additional FGF receptor was discovered, FGFR1, providing higher resolution on the process of specification (Kang et al. 2017). *Fgfr1* has a pan-ICM expression at early stages and is demonstrated to be the main responsible receptor for transducing the FGF4 signal by all ICM cells. Later expression of *Fgfr2* in only a subset of cells results in robust ERK activation via FGFR1 and FGFR2 and subsequent PE program activation (Kang et al. 2017). We thank the reviewer for his remark and we have updated this part of the discussion.

Line 383:

Interestingly, MAPK/Erk activation via FGF4 treatment of mouse blastocysts has been reported to induce pan-ICM PE fate^{4,23,24}. Several studies have corroborated the role of FGF/MAPK signaling in the resolution of the co-expression state and the EPI versus PE cell fate establishment but also in the maintenance of *Gata6* expression in the PE population^{23–25}. The prevailing model until recently posited the reciprocal expression of *Fgf4* and *Fgfr2* in EPI-biased and PE-biased cells respectively as the driving force of PE fate establishment^{24,91}. The latest model highlights the role of an additional FGF receptor, *Fgfr1*, which is expressed in all ICM cells at earlier stage than *Fgfr2* and can equally transduce the FGF4 signal⁹². Transduction of this signal through FGFR1 stimulates low Erk activity and facilitates the acquisition of the EPI fate by a subset of cells. Upon *Fgfr2* expression by the presumptive PE cells, robust Erk activation through both FGFR1 and FGFR2 leads to complete suppression of the EPI program, thereby activating the PE program. However, it has been proposed that this network of regulation is not solely centered on the repressive input of FGF/MAPK signaling on the EPI program but also involves an additional positive input of it onto GATA6 expression, directly activating the PE program⁹³.

-The discussion on the origin of Wnt ligands is unclear to me. To explain why the *Porcn*^{-/-} mutants do not have a pre-implantation phenotype, do the authors suggest that exogenous Wnt ligands, supplied by the maternal environment, are responsible for differentially activating Wnt in EPI and PrE cells? In this case, how would the authors explain that embryos cultured in minimal media (without any Wnt ligands) can specify the correct number of EPI and PrE cells?

The reviewer fairly raises an extremely interesting doubt regarding the origin of Wnt ligands and the correct specification of EPI and PE lineages in minimal media. Our article focuses on the external administration of DKK1 and how paracrine inhibition of canonical Wnt signaling affects lineage specification. Even though it is a very interesting question, investigating the origin of DKK1 or Wnt ligands was not part of our study. However, we tried to hypothesize and give a possible explanation in our discussion. It has been shown by various publications that Wnt components, including ligands, inhibitors, and receptors, are expressed in the early mouse embryo but also from the uterus environment, providing different potential sources of Wnt activating or inhibitory signals (Kemp et al. 2005, Tulac et al. 2003, Tepekoy et al. 2015, Harwood et al. 2009, Hayashi et al. 2009). The fact that *Porcn*^{-/-} do not have a pre-implantation defect favors the theory that Wnt signals may also be provided by the maternal tissues. However, the ability of the embryos to specify correct lineages *in vitro* can be attributed to their autonomous expression of Wnt signals. In addition, it has been previously demonstrated that embryos *in vitro* cultured in group develop better than single cultured embryos, meaning that paracrine signaling from one embryo to another can not be excluded either. The maternal- or embryo-originated signals theories are not conflicting since the two mechanisms can be complementary *in vivo* or redundant, as in the aforementioned cases. We have clarified this point in the discussion of the revised version.

Line 407:

It has been shown that Wnt agonists, antagonists and components involved in the transduction machinery are expressed in the mouse preimplantation embryo including *Wnt1*, *Wnt3a*, *Srpf1* and *Dkk1*^{29,99}. Interestingly, *Porcn*^{-/-} embryos, which are defective for Wnt secretion, did not show any defect on cell number or cell fate decisions in preimplantation development, indicating that autocrine embryo Wnt signaling activation is dispensable at these stages. In accordance, ligands and inhibitors of Wnt signaling are expressed in the murine and/or human endometrium in unique patterns at different stages of the menstrual cycle, indicative that embryo development does not solely rely on autologous Wnt signals^{53,100,101}.

Minor:

-Specification and commitment are used interchangeably in the manuscript.

We thank the reviewer for this comment. We think that “specification” is the most appropriate term in this context so we replaced “commitment” throughout the manuscript.

REVIEWERS' COMMENTS

Reviewer #1 (Remarks to the Author):

The authors have satisfactorily addressed all my concerns.

There is one sentence left on which I wish to draw their attention, at line 175.

In my opinion, at this stage of the narrative, the authors cannot conclude that the use of iCRT3 removes "the repressive transcriptional activity of TCF/LEF"; iCRT3 in fact essentially inhibits canonical TCF/LEF-beta-catenin activity. I suggest the authors to be clear and remove any ambiguity in the description of this assay.

Reviewer #2 (Remarks to the Author):

Reviewer #2 (Remarks to the Author):

Athanasouli et al. analyze the effects of Wnt signaling through Tcf7l1 during epiblast and primitive endoderm segregation in the pre-implantation embryo and during mESC differentiation. In contrast to a number of previous publications, they find that Wnt signaling does play a role in EPI/PrE segregation: active Wnt is characteristic of the EPI lineage and Tcf7l1 (a repressor, inactivated by Wnt) is needed in the PrE lineage.

In addition, using mESCs as an in vitro model they show that Tcf7l1 acts as an inhibitor of the pluripotency network and thereby promotes PrE fate.

Overall, there seems to be a minor change in Epi and PrE cell fates in the preimplantation embryo due to perturbations to Wnt/Tcf7l1 (Dkk1 treatment, iCRT3 treatment or Tcf7l1 knock-out). This may be in line with the subtle differences in beta-catenin intensities observed between NANOG and GATA6 positive cells in the embryo.

Main points:

-Trophoblast cells are not classified by staining for a specific marker. Lingering GATA6 expression in the

TE could therefore falsely be interpreted as an increase in PrE cells (a good example of this is the Dkk1- treated embryo in Figure 2c).

We agree with the reviewer and thank him/her for asking about this point.

We have stained 29 embryos (control) and 30 embryos (DKK1) from 3 independent experiments. Embryos have been treated with the WNT inhibitor DKK1 for 48h in culture. At E2.5+48h, embryos were immunostained for cell lineage- specific markers, CDX2 (TE), GATA6 (PE) and NANOG (EPI). Embryos were analyzed for GATA6 and CDX2 immuno- colocalization and only GATA6+/CDX2- /NANOG- were classified as PE (indicated as GATA6+ in the figure). NANOG+/GATA6- cells were classified as EPI (indicated as NANOG+ cells in the figure). CDX2+ were classified as TE. In accordance with our previous results, the new experiments and quantifications demonstrate that embryos treated with DKK1 show a significantly higher number of GATA6+ (PE) cells with a reduction of NANOG+ cell number indicating a lineage switch in the ICM. Interestingly, the number of CDX2+ cells did not change compared to the control (see new Fig. 2c-e).

Our results indicate that Wnt inhibition has the main effect on promoting PE formation. We have changed previous Fig. 2c-e, for new figures that include the CDX2 staining (see new figure below).

The text has been changed accordingly.

Line 152:

Lineages assessment showed that DKK1-treated embryos contained a significantly increased number

of PE (GATA6+) cells compared to control embryos. In addition, DKK1-treated embryos showed a strong but statistically non-significant tendency ($p=0.055$) to have fewer EPI cells (NANOG+) than untreated embryos. Interestingly, we did not observe a statistically significant difference in trophoctoderm (TE) cell number between control and treated embryos (Fig. 2c, d). Although the total number of ICM cells (Supplementary Fig. 2b) remained unchanged, DKK1-treated embryos exhibited an increased GATA6+ PE proportion and a decreased NANOG+ EPI proportion (Fig. 2e), demonstrating preferential PE specification at the expense of the EPI upon Wnt inhibition.

Fig. 2 Wnt signaling inhibition promotes formation of the PE lineage. a. Representative IF image of NANOG, GATA6 and CDX2 protein signals in E2.5+ 48H control and DKK1 treated embryo. Nuclei were counterstained with DAPI. Scale bar = 50 μ m. b. Number of NANOG+ ,GATA6+ and CDX2+ cells (counts) per embryo. Mean \pm SEM; Control: n=29; DKK1 n=30; 3 independent experiments. t test $**p<0.01$; ns= non significant. c. Percentage of NANOG+ and GATA6+ cells normalized on total number of ICM per embryo. Mean \pm SEM; Ctrl n=29, DKK1 n=30; 3 independent experiments.

Supplementary Fig. 2b: Total number of ICM cells per embryo. Mean \pm SEM; Control: n=29, DKK1: n=30; 3 independent experiments; unpaired t test; ns= non significant.

Thank you for including this experiment. Proper classification of TE cells did decrease the number of PrE cells previously reported and further underlines my previous statement that the effect of Wnt inhibition in vivo is minor on PrE fate. Currently I feel the effect is overstated in the text.
-The chimera experiment is not convincing at all. The signal looks very much like background, and there are no positive or negative controls. Using a cell line with a nuclear-localized fluorescent marker would help distinguish background vs real cell integration.

We agree with the reviewer that the chimera experiments need improvement. However, as answered to reviewer one, the facility of embryo injection has been experiencing several difficulties in obtaining chimera embryos (from our group and also other groups). Prior to blastocyst injection, the TCF7L1 OE cells need sorting and this creates an extra challenge to the experiment.
In accordance with the editor, as we can not provide new experiments in a short time, we have replaced the current images with the images that we had in the previous version of the article (Figure 4h,i). We have also introduced negative controls (Supplementary Fig. 4i).

Supplementary Fig. 4i: Representative BF and fluorescence image of E6.5 non-chimeric embryos injected with Tcf7l1 induced cells (Venushigh/dsRed+) (upper panel) and E6.5 non-injected embryos (lower panel).

Unfortunately, the new images provided are not convincing of in vivo chimeric potential of Tcf7l1 induced cells. Fluorescence seems more like cell debris. I sympathize with the difficulty of performing more experiments. I would recommend removing the chimera experiments altogether and stating that the in vivo potential of these cells remains to be determined.

-The role of Fgf4/Erk (which is well known for being involved in establishing EPI and PrE cell fates, and which when manipulated results in a complete change in cell fate in all ICM cells) is largely ignored in

this paper. How would Wnt/Tcf7l1 interact with the Fgf4/Erk pathway?

For example, if MAPK signaling is inhibited, could Wnt inhibition still drive PrE fate? Or conversely, if exogenous Fgf4 is added, could Tcf7l1 deletion still result in EPI cells? Would a Tcf7l1 pulse occur when Fgf4/Erk signaling is inhibited?

Based on the sentence below, do the authors propose that Wnt inhibition is responsible for the initiation of GATA6 expression in the early embryo?

"However, Fgf4^{-/-} embryos express Gata6, which cannot be maintained after E3.25. This suggests that the onset of the PE program is FGF4-independent." If so, this should be demonstrated by performing stainings for GATA6 in E3.25 embryos where Tcf7l1 is knocked out.

In summary, no clear model is proposed, nor tested for how the Wnt and MAPK pathways would interact during EPI/PrE fate specification.

FGF and Wnt signaling cooperate in several biological processes. As stated by the reviewer, FGF is the main signaling involved in establishing EPI and PE fates. It would be extremely interesting to investigate the relation/interaction between FGF and Wnt (TCF7L1) pathways. However, this would need several new experiments with a considerable delay in the article's publication. In accordance with the Journal, we have decided not to investigate this point in this article revision.

It is disappointing that these experiments were not attempted. These would not require the technical difficulty of generating chimeras.

-The change in lumen volume due to Wnt perturbations is interesting, however there is no evidence this has anything to do with the ICM. In fact, it is much more likely due to changes in the TE lineage. Here again, the manuscript would benefit from analysis of TE cell numbers and TE lineage specific marker expression.

The reference used for the sentence below shows that lumen formation facilitates the EPI/PrE fate decision, but does not state anything about cell fate having an effect on lumen formation. "A direct correlation between PE specification and blastocyst lumen expansion has been recently shown."

We thank the reviewer to point out this interesting suggestion. We have performed TE staining and quantification upon DKK1 treatment. Please, see answer to reviewer's first point. See new Figure 2c-e.

The reviewer is right. Our sentence "A direct correlation between PE specification and blastocyst lumen expansion has been recently shown." might induce some misunderstanding. We have changed the order of the sentence in the new version of the article.

Line 162:

A direct correlation between blastocyst lumen expansion and PE specification has been recently shown 59. Our results show that the blastocyst lumen volume of DKK1-treated embryos was 72% larger (Fig. 2f, g), which was associated with a significant increase of 80% in the embryo size area compared to control embryos (Fig. 2h, i and Supplementary Fig. 2d).

Ok.

-The sentence below has two inaccuracies. 1) Fgf4/Erk signaling is not regarded as a maintenance pathway for EPI and PrE cell fates. It is needed for the emergence of the salt-and-pepper NANOG and GATA6 expression pattern in the ICM from an initially co-expressing state. 2) Moreover, it is now well accepted that Fgfr2 is not the main receptor in the early embryo, but rather Fgfr1 is, which is expressed in all ICM cells. "In detail, FGF/MAPK is required for the maintenance of PE cells in the ICM as a result of the inverse expression of Fgf4/Fgfr2 in EPI-fated and PE-fated cells respectively."

1) We agree with the reviewer that FGF4/ERK signaling is not considered a maintenance pathway for both EPI and PE fates. As a matter of fact our initial sentence stated that "FGF/MAPK is required for the maintenance of PE cells in the ICM", which has been demonstrated by various articles. It has been shown that even though FGF4/ERK is not required to induce Gata6 expression, it is necessary for its maintenance in the PE-fated cells after the 32- cell stage (Kang et al., 2013; Krawchuk et al., 2013; Ohnishi et al., 2014). Therefore, we can't assess our statement as inaccurate. However, we appreciate that we neglected to mention the most recognized role of FGF4/ERK in the salt-and-pepper distribution of NANOG and GATA6 in the ICM. We have changed this part of the discussion in the new version. Line 383.

2) The reviewer is right. It was previously considered that EPI and PE fates within the ICM are established due to the inverse expression of FGF4/FGFR2. EPI-biased cells secrete FGF4, which is received by the PE-biased cells and transduced via the FGFR2 receptor, promoting paracrine induction of PE fate (Guo et al. 2010, Kang et al. 2013). While this is not inaccurate, it is partially incomplete since an additional FGF receptor was discovered, FGFR1, providing higher resolution on the process of specification (Kang et al. 2017). Fgfr1 has a pan-ICM expression at early stages and is demonstrated to be the main responsible receptor for transducing the FGF4 signal by all ICM cells. Later expression of Fgfr2 in only a subset of cells results in robust ERK activation via FGFR1 and FGFR2 and subsequent PE program activation (Kang et al. 2017). We thank the reviewer for his remark and we have updated this part of the discussion.

Line 383:

Interestingly, MAPK/Erk activation via FGF4 treatment of mouse blastocysts has been reported to induce pan-ICM PE fate 4,23,24. Several studies have corroborated the role of FGF/MAPK signaling in the resolution of the co- expression state and the EPI versus PE cell fate establishment but also in the maintenance of Gata6 expression in the PE population 23–25. The prevailing model until recently posited the reciprocal expression of Fgf4 and Fgfr2 in EPI- biased and PE-biased cells respectively as the driving force of PE fate establishment 24,91. The latest model highlights the role of an additional FGF receptor, Fgfr1, which is expressed in all ICM cells at earlier stage than Fgfr2 and can equally transduce the FGF4 signal 92. Transduction of this signal through FGFR1 stimulates low Erk activity and facilitates the acquisition of the EPI fate by a subset of cells. Upon Fgfr2 expression by the presumptive PE cells, robust Erk activation through both FGFR1 and FGFR2 leads to complete suppression of the EPI program, thereby activating the PE program. However, it has been proposed that this network of regulation is not solely centered on the repressive input of FGF/MAPK signaling on the EPI program but also involves an additional positive input of it onto GATA6 expression, directly activating the PE program 93.

Ok.

-The discussion on the origin of Wnt ligands is unclear to me. To explain why the *Porcn*^{-/-} mutants do not have a pre-implantation phenotype, do the authors suggest that exogenous Wnt ligands, supplied by the maternal environment, are responsible for differentially activating Wnt in EPI and PrE cells? In this case, how would the authors explain that embryos cultured in minimal media (without any Wnt ligands) can specify the correct number of EPI and PrE cells?

The reviewer fairly raises an extremely interesting doubt regarding the origin of Wnt ligands and the correct specification of EPI and PE lineages in minimal media. Our article focuses on the external administration of DKK1 and how paracrine inhibition of canonical Wnt signaling affects lineage specification. Even though it is a very interesting question, investigating the origin of DKK1 or Wnt ligands was not part of our study. However, we tried to hypothesize and give a possible explanation in our discussion. It has been shown by various publications that Wnt components, including ligands, inhibitors, and receptors, are expressed in the early mouse embryo but also from the uterus environment, providing different potential sources of Wnt activating or inhibitory signals (Kemp et al. 2005, Tulac et al. 2003, Tepekoy et al. 2015, Harwood et al. 2009, Hayashi et al. 2009). The fact that *Porcn*^{-/-} do not have a pre-implantation defect favors the theory that Wnt signals may also be provided by the maternal tissues. However, the ability of the embryos to specify correct lineages in vitro can be attributed to their autonomous expression of Wnt signals. In addition, it has been previously demonstrated that embryos in vitro cultured in group develop better than single cultured embryos, meaning that paracrine signaling from one embryo to another can not be excluded either. The maternal- or embryo- originated signals theories are not conflicting since the two mechanisms can be complementary in vivo or redundant, as in the aforementioned cases. We have clarified this point in the discussion of the revised version.

Line 407:

It has been shown that Wnt agonists, antagonists and components involved in the transduction machinery are expressed in the mouse preimplantation embryo including *Wnt1*, *Wnt3a*, *Srpf1* and *Dkk1* 29,99. Interestingly, *Porcn*^{-/-} embryos, which are defective for Wnt secretion, did not show any defect on cell number or cell fate decisions in preimplantation development, indicating that autocrine embryo Wnt signaling activation is dispensable at these stages. In accordance, ligands and inhibitors of Wnt signaling are expressed in the murine and/or human endometrium in unique patterns at different stages of the menstrual cycle, indicative that embryo development does not solely rely on autologous Wnt signals 53,100,101.

Ok.

REVIEWERS'**COMMENTS**

Reviewer #1 (Remarks to the Author):

The authors have satisfactorily addressed all my concerns.

There is one sentence left on which I wish to draw their attention, at line 175.

In my opinion, at this stage of the narrative, the authors cannot conclude that the use of iCRT3 removes “the repressive transcriptional activity of TCF/LEF”; iCRT3 in fact essentially inhibits canonical TCF/LEF-beta-catenin activity. I suggest the authors to be clear and remove any ambiguity in the description of this assay.

We agree with the reviewer that iCRT3 treatment does not remove repressive TCF/LEF transcriptional activity. Actually, iCRT3 removes the activating TCF/LEF transcriptional activity.

In the previous version of the article we explain that “iCRT3, which prevents the interaction of β -catenin and TCF factors, thus inhibiting transcription of Wnt target genes”. In addition, the use of iCRT3 allow us to conclude that “the repressive transcriptional activity of TCF/LEF is directly involved in EPI to PE specification”.

In summary, we do not conclude that iCRT3 removes the repressive transcriptional activity of TCF/LEF and therefore we have not proceeded to any text change.

Reviewer #2 (Remarks to the Author):

Athanasouli et al. analyze the effects of Wnt signaling through Tcf7l1 during epiblast and primitive endoderm segregation in the pre-implantation embryo and during mESC differentiation. In contrast to a number of previous publications, they find that Wnt signaling does play a role in EPI/PrE segregation: active Wnt is characteristic of the EPI lineage and Tcf7l1 (a repressor, inactivated by Wnt) is needed in the PrE lineage. In addition, using mESCs as an in vitro model they show that Tcf7l1 acts as an inhibitor of the pluripotency network and thereby promotes PrE fate.

Overall, there seems to be a minor change in Epi and PrE cell fates in the preimplantation embryo due to perturbations to Wnt/Tcf7l1 (Dkk1 treatment, iCRT3 treatment or Tcf7l1 knock-out). This may be in line with the subtle differences in beta-catenin intensities observed between NANOG and GATA6 positive cells in the embryo.

Main points:

-Trophectoderm cells are not classified by staining for a specific marker. Lingering GATA6 expression in the TE could therefore falsely be interpreted as an increase in PrE cells (a good example of this is the Dkk1- treated embryo in Figure 2c).

We agree with the reviewer and thank him/her for asking about this point. We have stained 29 embryos (control) and 30 embryos (DKK1) from 3 independent experiments. Embryos have been treated with the WNT inhibitor DKK1 for 48h in culture. At E2.5+48h, embryos were immunostained for cell lineage- specific markers, CDX2 (TE), GATA6 (PE) and NANOG (EPI). Embryos were analyzed for GATA6 and CDX2 immuno- colocalization and only GATA6+/CDX2-/NANOG- were classified as PE (indicated as GATA6+ in the figure). NANOG+/GATA6- cells were classified as EPI (indicated as NANOG+ cells in the figure). CDX2+ were classified as TE. In accordance with our previous results, the new experiments and quantifications demonstrate that embryos treated with DKK1 show a significantly higher number of GATA6+ (PE) cells with a reduction of NANOG+ cell number indicating a lineage switch in the ICM. Interestingly, the number of CDX2+ cells did not change compared to the control (see new Fig. 2c-e).

Our results indicate that Wnt inhibition has the main effect on promoting PE formation. We have changed previous Fig. 2c-e, for new figures that include the CDX2 staining (see new figure below). The text has been changed accordingly.

Line 152:

Lineages assessment showed that DKK1-treated embryos contained a significantly increased number of PE (GATA6+) cells compared to control embryos. In addition, DKK1-treated embryos showed a strong but

statistically non-significant tendency ($p=0.055$) to have fewer EPI cells (NANOG+) than untreated embryos. Interestingly, we did not observe a statistically significant difference in trophectoderm (TE) cell number between control and treated embryos (Fig. 2c, d). Although the total number of ICM cells (Supplementary Fig. 2b) remained unchanged, DKK1-treated embryos exhibited an increased GATA6+ PE proportion and a decreased NANOG+ EPI proportion (Fig. 2e), demonstrating preferential PE specification at the expense of the EPI upon Wnt inhibition.

Fig. 2 Wnt signaling inhibition promotes formation of the PE lineage. a. Representative IF image of NANOG, GATA6 and CDX2 protein signals in E2.5+ 48H control and DKK1 treated embryo. Nuclei were counterstained with DAPI. Scale bar = 50 μ M. b. Number of NANOG+ ,GATA6+ and CDX2+ cells (counts) per embryo. Mean \pm SEM; Control: n=29; DKK1 n=30; 3 independent experiments. t test ** $p<0.01$; ns= non significant. c. Percentage of NANOG+ and GATA6+ cells normalized on total number of ICM per embryo. Mean \pm SEM; Ctrl n=29, DKK1 n=30; 3 independent experiments.

Supplementary Fig. 2b: Total number of ICM cells per embryo. Mean \pm SEM; Control: n=29, DKK1: n=30; 3 independent experiments; unpaired t test; ns= non significant.

Thank you for including this experiment. Proper classification of TE cells did decrease the number of PrE cells previously reported and further underlines my previous statement that the effect of Wnt inhibition in vivo is minor on PrE fate. Currently I feel the effect is overstated in the text.

We thank the reviewer for proposing this experiment as we believe that it has improved the conclusions of the article. We see that inhibition of Wnt pathway, by DKK1 treatment, has a significant effect mainly to the PrE cells and not on TE cells. We have been checking the text to review whether we were overstating our results. We believe that we have been careful to indicate only when results were significant without overstating it.

-The chimera experiment is not convincing at all. The signal looks very much like background, and there are no positive or negative controls. Using a cell line with a nuclear-localized fluorescent marker would help distinguish background vs real cell integration.

We agree with the reviewer that the chimera experiments need improvement. However, as answered to reviewer one, the facility of embryo injection has been experiencing several difficulties in obtaining chimera embryos (from our group and also other groups). Prior to blastocyst injection, the TCF7L1 OE cells need sorting and this creates an extra challenge to the experiment.

In accordance with the editor, as we can not provide new experiments in a short time, we have replaced the current images with the images that we had in the previous version of the article (Figure 4h,i). We have also introduced negative controls (Supplementary Fig. 4i).

Supplementary Fig. 4i: Representative BF and fluorescence image of E6.5 non-chimeric embryos injected with Tcf7l1 induced cells (Venushigh/dsRed+) (upper panel) and E6.5 non-injected embryos (lower panel).

Unfortunately, the new images provided are not convincing of in vivo chimeric potential of Tcf7l1 induced cells. Fluorescence seems more like cell debris. I sympathize with the difficulty of performing more experiments. I would recommend removing the chimera experiments altogether and stating that the in vivo potential of these cells remains to be determined.

We agree with the reviewer that our chimera experiments need improvement. We thank the reviewer for his/her understanding as this experiment would take a long time due to some technical problems. We decided to follow the reviewer's suggestion. Chimera's experiments have been removed from the figures and the text changed accordingly. We have added the statement that in vivo potential of TCF7L1 OE cells remains to be determined as suggested by the reviewer.

Line 243-246:

“Overall, these findings demonstrate that upon forced expression of *Tcf7l1*, cells engage in PE gene activation, revealing the key role played by TCF7L1 in PE differentiation of mESCs. Whether these cells can contribute to the extraembryonic layers of mouse embryos still needs to be examined.”

-The role of Fgf4/Erk (which is well known for being involved in establishing EPI and PrE cell fates, and which

when manipulated results in a complete change in cell fate in all ICM cells) is largely ignored in this paper. How would Wnt/Tcf711 interact with the Fgf4/Erk pathway?

For example, if MAPK signaling is inhibited, could Wnt inhibition still drive PrE fate? Or conversely, if exogenous Fgf4 is added, could Tcf711 deletion still result in EPI cells? Would a Tcf711 pulse occur when Fgf4/Erk signaling is inhibited?

Based on the sentence below, do the authors propose that Wnt inhibition is responsible for the initiation of GATA6 expression in the early embryo?

“However, Fgf4^{-/-} embryos express Gata6, which cannot be maintained after E3.25. This suggests that the onset of the PE program is FGF4-independent.” If so, this should be demonstrated by performing stainings for GATA6 in E3.25 embryos where Tcf711 is knocked out.

In summary, no clear model is proposed, nor tested for how the Wnt and MAPK pathways would interact during EPI/PrE fate specification.

FGF and Wnt signaling cooperate in several biological processes. As stated by the reviewer, FGF is the main signaling involved in establishing EPI and PE fates. It would be extremely interesting to investigate the relation/interaction between FGF and Wnt (TCF7L1) pathways. However, this would need several new experiments with a considerable delay in the article's publication. In accordance with the Journal, we have decided not to investigate this point in this article revision.

It is disappointing that these experiments were not attempted. These would not require the technical difficulty of generating chimeras.

We agree that the reviewer has pointed an interesting point that we aim to pursue in the future. However, in agreement with the Journal we decided not to perform these experiments in the current version of the article.

-The change in lumen volume due to Wnt perturbations is interesting, however there is no evidence this has anything to do with the ICM. In fact, it is much more likely due to changes in the TE lineage. Here again, the manuscript would benefit from analysis of TE cell numbers and TE lineage specific marker expression. The reference used for the sentence below shows that lumen formation facilitates the EPI/PrE fate decision, but does not state anything about cell fate having an effect on lumen formation. “A direct correlation between PE specification and blastocyst lumen expansion has been recently shown.”

We thank the reviewer to point out this interesting suggestion. We have performed TE staining and quantification upon DKK1 treatment. Please, see answer to reviewer's first point. See new Figure 2c-e.

The reviewer is right. Our sentence “A direct correlation between PE specification and blastocyst lumen expansion has been recently shown.” might induce some misunderstanding. We have changed the order of the sentence in the new version of the article.

Line 162:

A direct correlation between blastocyst lumen expansion and PE specification has been recently shown 59. Our results show that the blastocyst lumen volume of DKK1-treated embryos was 72% larger (Fig. 2f, g), which was associated with a significant increase of 80% in the embryo size area compared to control embryos (Fig. 2h, i and Supplementary Fig. 2d).

Ok.

We are pleased that the reviewer agrees with the changes.

-The sentence below has two inaccuracies. 1) Fgf4/Erk signaling is not regarded as a maintenance pathway for EPI and PrE cell fates. It is needed for the emergence of the salt-and-pepper NANOG and GATA6 expression pattern in the ICM from an initially co-expressing state. 2) Moreover, it is now well accepted that Fgfr2 is not the main receptor in the early embryo, but rather Fgfr1 is, which is expressed in all ICM cells. “In detail,

FGF/MAPK is required for the maintenance of PE cells in the ICM as a result of the inverse expression of Fgf4/Fgfr2 in EPI-fated and PE-fated cells respectively.”

1) We agree with the reviewer that FGF4/ERK signaling is not considered a maintenance pathway for both EPI and PE fates. As a matter of fact our initial sentence stated that “FGF/MAPK is required for the maintenance of PE cells in the ICM”, which has been demonstrated by various articles. It has been shown that even though FGF4/ERK is not required to induce Gata6 expression, it is necessary for its maintenance in the PE-fated cells after the 32- cell stage (Kang et al., 2013; Krawchuk et al., 2013; Ohnishi et al., 2014). Therefore, we can't assess our statement as inaccurate. However, we appreciate that we neglected to mention the most recognized role of FGF4/ERK in the salt-and-pepper distribution of NANOG and GATA6 in the ICM. We have changed this part of the discussion in the new version. Line 383.

2) The reviewer is right. It was previously considered that EPI and PE fates within the ICM are established due to the inverse expression of FGF4/FGFR2. EPI-biased cells secrete FGF4, which is received by the PE-biased cells and transduced via the FGFR2 receptor, promoting paracrine induction of PE fate (Guo et al. 2010, Kang et al. 2013). While this is not inaccurate, it is partially incomplete since an additional FGF receptor was discovered, FGFR1, providing higher resolution on the process of specification (Kang et al. 2017). Fgfr1 has a pan-ICM expression at early stages and is demonstrated to be the main responsible receptor for transducing the FGF4 signal by all ICM cells. Later expression of Fgfr2 in only a subset of cells results in robust ERK activation via FGFR1 and FGFR2 and subsequent PE program activation (Kang et al. 2017). We thank the reviewer for his remark and we have updated this part of the discussion.

Line 383:

Interestingly, MAPK/Erk activation via FGF4 treatment of mouse blastocysts has been reported to induce pan-ICM PE fate 4,23,24. Several studies have corroborated the role of FGF/MAPK signaling in the resolution of the co-expression state and the EPI versus PE cell fate establishment but also in the maintenance of Gata6 expression in the PE population 23–25. The prevailing model until recently posited the reciprocal expression of Fgf4 and Fgfr2 in EPI- biased and PE-biased cells respectively as the driving force of PE fate establishment 24,91. The latest model highlights the role of an additional FGF receptor, Fgfr1, which is expressed in all ICM cells at earlier stage than Fgfr2 and can equally transduce the FGF4 signal 92. Transduction of this signal through FGFR1 stimulates low Erk activity and facilitates the acquisition of the EPI fate by a subset of cells. Upon Fgfr2 expression by the presumptive PE cells, robust Erk activation through both FGFR1 and FGFR2 leads to complete suppression of the EPI program, thereby activating the PE program. However, it has been proposed that this network of regulation is not solely centered on the repressive input of FGF/MAPK signaling on the EPI program but also involves an additional positive input of it onto GATA6 expression, directly activating the PE program 93.

Ok.

We are pleased that the reviewer agrees with the changes.

-The discussion on the origin of Wnt ligands is unclear to me. To explain why the *Porcn*^{-/-} mutants do not have a pre-implantation phenotype, do the authors suggest that exogenous Wnt ligands, supplied by the maternal environment, are responsible for differentially activating Wnt in EPI and PrE cells? In this case, how would the authors explain that embryos cultured in minimal media (without any Wnt ligands) can specify the correct number of EPI and PrE cells?

The reviewer fairly raises an extremely interesting doubt regarding the origin of Wnt ligands and the correct specification of EPI and PE lineages in minimal media. Our article focuses on the external administration of DKK1 and how paracrine inhibition of canonical Wnt signaling affects lineage specification. Even though it is a very interesting question, investigating the origin of DKK1 or Wnt ligands was not part of our study. However, we tried to hypothesize and give a possible explanation in our discussion. It has been shown by various publications that Wnt components, including ligands, inhibitors, and receptors, are expressed in the early mouse embryo but also from the uterus environment, providing different potential sources of Wnt activating or inhibitory signals (Kemp et al. 2005, Tulac et al. 2003, Tepekoy et al. 2015, Harwood et al. 2009, Hayashi et al. 2009). The fact that *Porcn*^{-/-} do not have a pre-implantation defect favors the theory that Wnt signals may also be provided by the maternal tissues. However, the ability of the embryos to specify correct lineages in vitro can be attributed to their autonomous expression of Wnt signals. In addition, it has been previously demonstrated that embryos in vitro cultured in group develop better than single cultured embryos, meaning that paracrine signaling from one embryo to another can not be excluded either. The maternal- or embryo-

originated signals theories are not conflicting since the two mechanisms can be complementary in vivo or redundant, as in the aforementioned cases. We have clarified this point in the discussion of the revised version.

Line 407:

It has been shown that Wnt agonists, antagonists and components involved in the transduction machinery are expressed in the mouse preimplantation embryo including Wnt1, Wnt3a, Srfp1 and Dkk1 29,99. Interestingly, Porcn^{-/-} embryos, which are defective for Wnt secretion, did not show any defect on cell number or cell fate decisions in preimplantation development, indicating that autocrine embryo Wnt signaling activation is dispensable at these stages. In accordance, ligands and inhibitors of Wnt signaling are expressed in the murine and/or human endometrium in unique patterns at different stages of the menstrual cycle, indicative that embryo development does not solely rely on autologous Wnt signals 53,100,101.

Ok.

We are pleased that the reviewer agrees with the comment/changes.